# Bridging non-overlapping reads illuminates high-order epistasis between distal protein sites in a GPCR

Justin I. Yoo[1], Patrick S. Daugherty[1,2] & Michelle A. O'Malley [1]*

Epistasis emerges when the effects of an amino acid depend on the identities of interacting residues. This phenomenon shapes fitness landscapes, which have the power to reveal evolutionary paths and inform evolution of desired functions. However, there is a need for easily implemented, high-throughput methods to capture epistasis particularly at distal sites. Here, we combine deep mutational scanning (DMS) with a straightforward data processing step to bridge reads in distal sites within genes (BRIDGE). We use BRIDGE, which matches non-overlapping reads to their cognate templates, to uncover prevalent epistasis within the binding pocket of a human G protein-coupled receptor (GPCR) yielding variants with 4-fold greater affinity to a target ligand. The greatest functional improvements in our screen result from distal substitutions and substitutions that are deleterious alone. Our results corroborate findings of mutational tolerance in GPCRs, even in conserved motifs, but reveal inherent constraints restricting tolerated substitutions due to epistasis.

[1] Department of Chemical Engineering, University of California Santa Barbara, Santa Barbara, CA 93106, USA. [2] Serimmune, 150 Castillian Dr., Goleta, CA 93117, USA. *email: momalley@ucsb.edu

The directed evolution of proteins draws inspiration from nature to generate novel or improved phenotypes suited to a wide variety of biotechnological applications. Using iterative mutagenesis and selection, a protein's sequence is altered in a "walk" through sequence space in search of variants exhibiting a desired function[1]. Developed over the past decade, deep mutational scanning[2] combines next-generation sequencing (NGS) and protein engineering to construct expansive sequence–function landscapes and rapidly identify fitness optima, turning the "walk" through sequence space into a satellite survey. Information-rich landscapes preclude pitfalls associated with the iterative combination of beneficial mutations, which are often unpredictable[3,4]. For example, evolutionary trajectories can be constrained to local fitness optima, and mutations are often deleterious in isolation but beneficial in a different sequence context, a phenomenon known as sign epistasis[4–7]. Accurate construction of these landscapes requires identification of all mutations in each protein variant due to the unpredictable effects of epistasis[8].

DMS analyses use overlapping paired-end reads to reduce error rates by sequencing each DNA segment twice[9]. However, the need for overlapping reads and read-length limitations restrict identification of concurrent substitutions to protein libraries containing proximal mutations (i.e., all mutations located within less than one read length, <100 amino acids using Illumina NextSeq and HiSeq). Clever solutions to overcome read-length limitations have been developed[10–15], and several have been applied to DMS[15–17]; however, these methods are limited by their time- and work-intensive nature, throughput, and fidelity. For example, subassembly has been extended to DMS through multiple en masse digestion and intramolecular ligation reactions of gene libraries to reposition the coding sequence proximal to a molecular barcode within the vector backbone[16,17]. Dilute concentrations of DNA are required to minimize intermolecular interactions, and each reaction introduces the risk of masking the enrichment of variants with high fitness due to unsuccessful DNA manipulation. Moreover, the incorporation of barcodes in the backbone required construction and amplification of a separate library in E. coli[16], limiting the size of a Gal4 library to minimize association of a distinct barcode to multiple inserts. Introduction of barcodes through microfluidics[11] necessitates specialized instrumentation and expertise, and emulsion PCR[9] is associated with polyclonal barcoding and inefficient amplification of multiple loci. Thus, there is a critical need to develop accessible methods to fully explore the fitness landscapes of key residues that are distal in sequence, yet important to protein function, as is common to sites involved in ligand binding or biocatalysis (Fig. 1a).

To address these needs, we apply a simple data-processing step inspired by genome assembly algorithms that leverages the Illumina sequencing by synthesis (SBS) mechanism to match short, nonoverlapping paired-end reads located distally within a gene. In this work, we demonstrate the preservation of nonoverlapping read pairs for DMS to account for epistasis at distal positions. In addition, we present an NGS design consideration that results in single read error rates between 0.02 and 0.05%, which are comparable with those sought using overlapping paired-end sequencing. As a proof-of-principle, we chose to engineer a model GPCR, the human adenosine $A_2a$ receptor ($A_2aR$), as critical residues in GPCR binding pockets are spread across multiple transmembrane domains exceeding the longest read lengths offered by Illumina SBS platforms. Further, this protein family is important in human physiology[18,19], drug targeting[20], and its capacity for biosensing[21]. Accordingly, we combine DMS with BRIDGE to engineer $A_2aR$ ligand binding, a phenotype with biotechnological and therapeutic relevance, directly taking into account the effects of high-order epistasis. BRIDGE is compatible with any DMS experiment utilizing Illumina paired-end NGS, and does not require any specialized materials or additional experimental steps.

## Results

**Development and optimization of FACS-based GPCR screen.** In this study, we sought to improve the binding affinity of $A_2aR$ toward FITC-APEC[22], a fluorescent adenosine receptor agonist. This choice was motivated by the structural similarity between the ligand moiety and the agonist NECA, which is bound by $A_2aR$ in several crystal structures[23–25] providing molecular insight for library design and downstream analysis. To enable DMS of a GPCR library, we first developed a fluorescence-activated cell sorting (FACS)-based screen using a fluorescent ligand to couple cellular fluorescence intensity to ligand binding (Fig. 2). We initially sought to improve the sensitivity of the FACS-based screen through optimization of GPCR production and fluorescent ligand binding in cells producing wild-type $A_2aR$. Fluorescein has greatest fluorescence intensity in its dianionic form[26]; therefore, ligand binding and FACS analyses were performed in buffer at least 1 pH unit above the pKa of the dianion (6.43). We previously demonstrated that the expression vector backbone can significantly affect the yield of the total and functional $A_2aR$ in Saccharomyces cerevisiae[27]. In agreement with previous results using GFP-tagged $A_2aR$[27], expression of decahistidine-tagged $A_2aR$ from a multi-site-integrating backbone improves specific signal 11-fold compared with a centromeric vector upon incubation with 10 μM FITC-APEC (Supplementary Fig. 1). As the multi-site integrating vector provides the greatest dynamic range in fluorescent signal and, likely, sensitivity to changes in experimental parameters, this system was used for subsequent optimization of GPCR production and ligand binding.

While screening a protein library, it is ideal to represent all variants equally. However, phenotypic heterogeneity arises often due to differences in protein production, which can introduce bias while screening libraries. Indeed, phenotypic heterogeneity emerged in previous GPCR engineering efforts upon fluorescent ligand binding and agonist-induced GFP expression[28,29]. Upon expression of $A_2aR$ and incubation with fluorescent ligand, we also observe heterogeneity of functional protein production manifesting in a bimodal fluorescent population where 10% of the population exhibits fluorescence intensities comparable with autofluorescence (Supplementary Fig. 2b, c). Here, $A_2aR$ expression is under the control of a galactose-inducible promoter ($P_{GAL1}$) (Supplementary Fig. 2a), as inducible promoter systems are ideal for the expression of heterologous GPCRs, which often impart cytotoxic effects on yeast cells[30]. Culturing cells with a neutral carbon source (e.g., lactose, raffinose, etc.), neither inducing nor repressing $P_{GAL1}$-driven gene expression, prior to induction of gene expression has been implemented to prepare the cell for growth and gene expression in galactose, as several genes are regulated by this catabolite[31,32]. Upon introduction of a similar pre-induction step using raffinose as a neutral carbon source, we observe a phenotypically homogeneous cell population upon incubation with FITC-APEC (Supplementary Fig. 2b, c). The change from a bimodal distribution to a unimodal distribution is critical during library screening so that rare, gain-of-function mutants are not excluded due to expression biases. The improvement in phenotypic homogeneity is concomitant with a minimal improvement in mean fluorescence intensity (MFI) over background (Supplementary Fig. 2d). Upon optimization of GPCR expression and FACS parameters, only two rounds of sorting are required to fully enrich yeast cells producing wild-type $A_2aR$ diluted (0.1%) into a pool of inactive

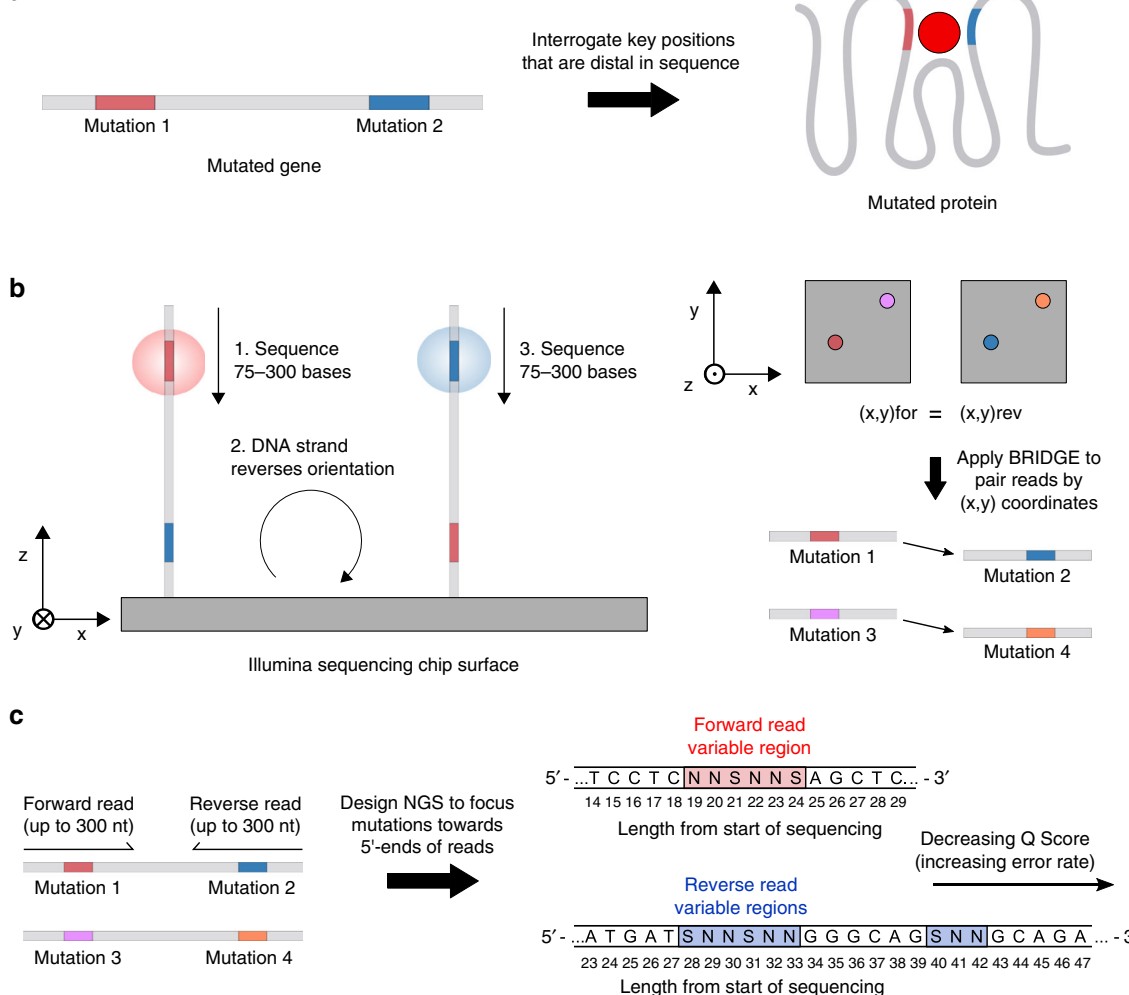

**Fig. 1 Diagram of Illumina sequencing mechanism underlying BRIDGE. a** New methods are needed to engineer key residues in active sites (e.g., ligand-binding domains) while accounting for epistatic interactions. **b** BRIDGE can be used to extensively quantify the enrichment of variants containing mutations at distal sites. In order to match nonoverlapping paired-end reads, we leverage the Illumina sequencing mechanism wherein reads generated from either end of the same DNA strand produce fluorescent signals at the same (x, y) coordinate on the sequencing chip surface. By matching nonoverlapping read pairs using their surface coordinates, BRIDGE directly accounts for epistasis between distal residues in experiments reliant on next-generation sequencing. **c** High sequencing accuracy is obtained by designing NGS experiments such that variable regions are focused towards the 5'-end of each read, where error rates are low.

variants (Supplementary Fig. 3). Further, saturation binding experiments demonstrate our ability to use fluorescent ligand binding to discriminate between active GPCR variants based on binding affinity (Supplementary Fig. 4).

Notably, the binding affinity for FITC-APEC is significantly lower than that measured for $A_2aR$ suspended in bovine striatal membranes[22]. The yeast cell wall has been reported to hinder transport of fluorescent ligands to the plasma membrane, necessitating permeabilization to facilitate saturation of membrane-embedded receptors[28]. Our results suggest that the use of high FITC-APEC concentrations leads to effective saturation without cell-wall permeabilization. Further, saturation promotes enrichment of variants with improved functional yields; thus, cell-wall modification is unnecessary to provide conditions favoring enrichment of variants with improved binding affinities.

**Structural analysis guides $A_2aR$ library design**. To identify $A_2aR$ residues as candidates for mutation, we used UCSF Chimera[33] to quantify atomic interactions between residues and bound ligands

in several crystal structures (Supplementary Table 1). These results are in agreement with putative conserved residue–ligand interactions, as well as a similar bioinformatics effort reported in the literature[23,34]. To refine our list of candidate residues, we identified previously reported single amino acid substitutions and their concomitant changes in binding affinity (Supplementary Table 2). Mutations conferring significant gain- or loss-of-function phenotypes toward NECA were sought as internal controls for FACS-based screening. Taken together, the lists of atomic contacts and previously reported mutations were used to identify five residues (T88[3.36], Q89[3.37], W246[6.48], L249[6.51], and H250[6.52]) (Supplementary Fig. 5) for site-saturation mutagenesis to test our GPCR screening pipeline. Here, the Ballesteros–Weinstein notation[35] is used in superscript to represent the relative position of a residue with respect to the most evolutionarily conserved residue in a transmembrane (TM) helix. The codon degeneracy used to construct the library (NNS) includes a stop codon, which acts as an additional negative control since premature termination at any of these positions will lead to complete loss-of-function.

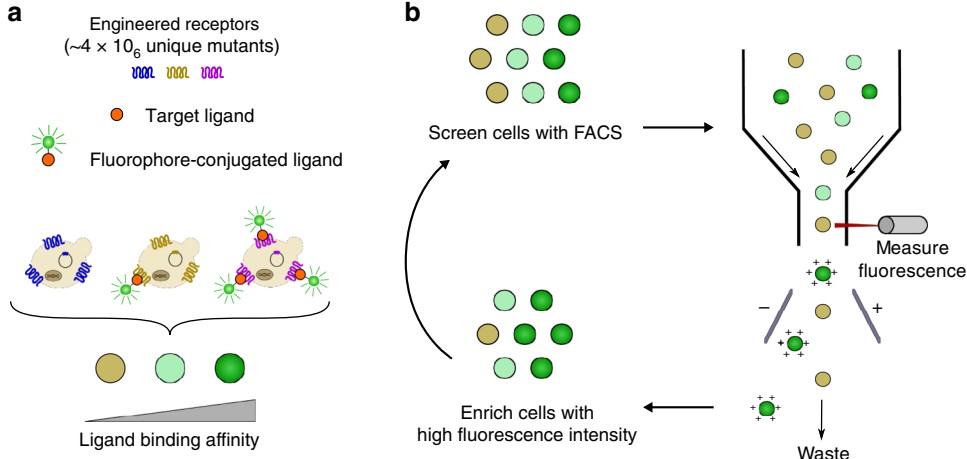

**Fig. 2 Diagram of FACS-based GPCR directed evolution and screen. a** A library of the human adenosine $A_2a$ receptor ($A_2aR$) was generated using degenerate NNS codons to perform saturation mutagenesis at five sites within the ligand-binding pocket. Yeast are transformed with the $A_2aR$ library and incubated with fluorescent ligand to couple ligand-binding affinity to cellular fluorescence intensity. **b** The yeast library is screened using fluorescence-activated cell sorting (FACS) to enrich cells producing $A_2aR$ variants with high fluorescence intensities. This process is repeated to further enrich receptors with improved ligand-binding affinities while depleting the population of variants with low affinities.

**FACS screen enriches variants improved in binding or yield**. Upon transforming the library into yeast, cells were incubated with a near-saturating concentration of FITC-APEC and subjected to four rounds of FACS-based screening. In the third and fourth rounds, we implemented high-stringency gating wherein we sorted the top 5% of cells with respect to mean fluorescence intensity (MFI) (Supplementary Fig. 6). After four rounds of screening, we isolated a population exhibiting a 3.5-fold improvement in MFI over wild-type $A_2aR$ (Supplementary Fig. 7), suggesting enrichment of variants with improved ligand-binding affinity and/or functional yield. In parallel, we sorted the library through four rounds of FACS using low-stringency gating, sorting all cells displaying MFI above background auto-fluorescence (Supplementary Fig. 6). Use of this approach significantly diminished the sequential improvement in cellular MFI obtained using high-stringency gating (Supplementary Fig. 7). As our goal is to identify variants with the greatest improvements in ligand-binding affinity, subsequent analyses focus on characterizing the libraries isolated using high-stringency gating. We chose to use a near-saturating ligand concentration for library screening after taking into consideration the rarity of beneficial mutations. We reasoned that this strategy would enable enrichment of variants with gains in functional yield, which would provide useful data, if variants with gains in binding affinity are scarce.

**BRIDGE matches nonoverlapping reads to cognate DNA templates**. In order to apply DMS to a library containing mutations in distal sites while preserving read pairs, we implemented a straightforward data-processing step to couple reads originating from the same DNA strand irrespective of their positions within the sequence. This processing step leverages the bridge amplification mechanism utilized during paired-end sequencing for Illumina SBS platforms[36] (see detailed explanation of BRIDGE in Supplementary Note 1). In short, the fluorescent signals generated upon sequencing either paired-end read from a single DNA strand occur in sufficient proximity on the sequencing chip surface such that they share a single $(x, y)$ coordinate (Fig. 1b). Using this strategy, we performed next-generation sequencing of barcoded $A_2aR$ libraries and developed custom python code to scan all sequencing data, quality-filter, trim, and append the reverse complement of each reverse read to the forward read with the same $(x, y)$ coordinate. Although the

mutated sites in our library occupy two distal loci, BRIDGE can be applied to libraries containing mutations in multiple distal sites (Supplementary Note 2). After processing the reads, we observe $\sim 1.7 \times 10^6$ unique protein variants in the naive library, $\sim 42\%$ of the theoretical diversity including stop codons.

**Design of NGS primers markedly reduces error rates**. Illumina NGS error rates approach $\sim 1\%$[37] (i.e., 1 error in every 100 bases); thus, DMS is often restricted to proximal mutations to enable overlapping paired-end sequencing, increasing the likelihood of accurate base calls. For example, if a DNA segment is sequenced twice using overlapping reads, the error rate associated with the overlapping segment decreases to $\sim 0.01\%$. However, errors are more prevalent at the 3′-end of Illumina sequencing reads[38]. Accordingly, by designing an NGS experiment to group variable regions near the 5′-ends of the forward and reverse reads, we hypothesized that we could significantly reduce the error rates associated with using single reads for each mutated locus (Fig. 1c). Indeed, we observe an average quality (Q) score of $\sim 35$ for the mutated bases in both forward and reverse reads, corresponding to a predicted error rate of 0.03% (Supplementary Table 3). Q scores are often overestimates of the true error rate; thus, to validate this prediction, we calculated the average Q scores, predicted error rates, and observed error rates associated with unaltered bases flanking the mutated regions in the forward and reverse reads. These base calls are also highly accurate with an average predicted error rate of 0.03% and observed error rates between $\sim 0.02$ and 0.05%, slightly greater than the value sought using overlapping paired-end reads (Supplementary Table 4).

**Enrichment rates of internal controls match expectations**. To assess the efficacy of the screen and data analysis pipeline, we first quantified the enrichment or depletion of internal controls from the naive to the post sort (PS) 4 population. Calculation of enrichment and depletion rates was performed, excluding variants with abundance below 15 in the naive library to minimize noise. In all but one case, the behavior of the controls matched expectations (Supplementary Table 2). Specifically, beneficial mutations, H250N[6.52] and Q89A/D[3.37], were enriched 4-, 149-, and 0.73-fold, respectively, exceeding the enrichment rate of wild-type $A_2aR$ (0.30-fold). In contrast, deleterious mutations (T88D/E/R/S[3.36], W246A[6.48], and H250F/Y[6.52]) were either highly

depleted or not detected in the PS4 population. The behavior of T88$^{3.36}$ was the only example in disagreement with previous reports[39,40], exhibiting an enrichment rate (0.49) slightly greater than that of the wild-type receptor. As expected, variants containing stop codons were significantly depleted, ~1000-fold in our screen, decreasing from 13.5% of the total reads in the naive library to 0.01% of the PS4 population (Supplementary Fig. 8). The estimated nonspecific carryover rate for each sort round is <1 (0.01–0.70), reflecting continuous depletion of stop codon variants and a stringent screen (Supplementary Fig. 8). At the nucleotide level, enrichment rate is not correlated with codon adaptation index (CAI)[41], indicating codon bias due to translational selection does not influence enrichment in our screen (Supplementary Fig. 9).

**Epistasis strongly influences enrichment of A$_2$aR variants**. The stringency of our screen is also evident in the preferential enrichment of few unique variants with, presumably, improved binding or expression phenotypes (Fig. 3a, Table 1). Strikingly, variants containing mutations in both distal sites exhibit up to 2.4-, 5.3-, and 10.2-fold greater enrichment rates compared with the most highly enriched variants with single, proximal double, or proximal triple mutations, respectively, which are amenable to conventional DMS analysis (Fig. 3b, Table 1). Moreover, among the 20 most highly enriched variants containing mutations in distal sites, 13 contain at least 1 deleterious single substitution reflecting sign epistasis (Fig. 3c).

Despite high conservation at the mutated sites in A$_2$aR orthologs (Supplementary Fig. 10), functionally improved variants (enrichment rate > 0.30) exhibit strong amino acid preferences at only two positions (Supplementary Fig. 11a). Threonine and serine are most frequently observed at T88$^{3.36}$, and leucine is most common at L249$^{6.51}$. The preference for the wild-type residue at T88$^{3.36}$ becomes greater after restricting the frequency analysis to variants with enrichment rates >61, representing the 95th percentile and variants likely favoring an active conformation (Supplementary Fig. 11b). A preference for the wild-type residue or serine at T88$^{3.36}$ is not surprising as this residue establishes contacts between agonists and transmembrane (TM) helix 3 in all agonist-bound A$_2$aR crystal structures[23,25,42,43], suggesting a key role in agonist binding. In contrast, Q89$^{3.37}$, W246$^{6.48}$, and H250$^{6.52}$ are permissive and sample several residues with varying side-chain chemistries (Supplementary Fig. 11a, b). The tolerance at position W246$^{6.48}$ is notable due to its presence in the highly conserved CWxP motif in class A GPCRs[44] and its putative role in A$_2$aR activation as a rotamer toggle switch[45] or transmission switch[46]. The tolerance to substitutions observed in this data set corroborates a previous DMS study wherein another class A GPCR, the rat neurotensin receptor 1, was interrogated at positions 43–418 using single-site-saturation mutagenesis[47]. It is important to note that while a diverse set of substitutions retain or improve agonist binding, epistasis constrains tolerated substitutions to those present alongside specific sequence backgrounds. As a result, few unique variants exhibit enrichment rates above that of wild-type (n = 740) or above 61 (n = 37) (Supplementary Fig. 11a, b). Mapping these fitness optima onto a sequence–function landscape would result in few, distant peaks reflecting the difficulty of identifying the rare sequences that confer significant gains in function.

**Radioligand binding confirms enhanced binding affinities**. To verify that enriched variants exhibit improved binding affinities toward our target ligand, binding affinities were determined using saturation radioligand binding to $^3$H-NECA. Importantly, whole-cell saturation binding enables quantification of functional

protein yield ($B_{max}$), which can, in theory, affect a variant's enrichment in our screen. As an additional means of analyzing variant enrichment, the processed reads from PS1 to PS4 were rank-ordered using Enrich2[48] (Supplementary Table 5), a software package commonly used for DMS. Variant GRIA (T88$^{3.36}$/Q89G$^{3.37}$/W246R$^{6.48}$/L249I$^{6.51}$/H250A$^{6.52}$) was ranked highly in both PS4/naive enrichment rate and Enrich2 scoring; thus, we chose this variant as well as variant SLNIG for determination of ligand-binding affinity and functional yield. In addition to variants GRIA and SLNIG, we chose to characterize the most abundant variant in PS4 (Q89S$^{3.37}$) and the positive control Q89A$^{3.37}$, which were enriched 336 and 149-fold, respectively (Fig. 3a, Table 1). In line with our goal, all assayed variants bind $^3$H-NECA with approximately fourfold greater affinities than the wild-type receptor, which exhibits an equilibrium dissociation constant ($K_d$) of 39.2 ± 4.8 nM in agreement with a previous report[49] (Fig. 4). The highly enriched variants, GRIA ($K_d$ of 10.9 ± 0.8 nM) and SLNIG ($K_d$ of 10.7 ± 0.6 nM), bind the target ligand with affinities similar to those of variants Q89A$^{3.37}$ ($K_d$ of 9.9 ± 0.4 nM) and Q89S$^{3.37}$ ($K_d$ of 11.8 ± 1.0 nM). Although these results suggest that differences in enrichment rates do not necessarily correlate with changes in ligand-binding affinity, improvements in fluorescent ligand binding may not directly translate to the binding of $^3$H-NECA. Nevertheless, BRIDGE enables identification of highly improved variants subject to high-order epistasis between distal sites. Variants GRIA and SLNIG represent sequences that would be difficult to discover using conventional methods; thus, their identification and inclusion in functional receptor data sets will be useful in informing studies on GPCR structure–function relationships.

Each of the highly enriched variants demonstrate greater improvements, over the wild-type receptor, in ligand-binding affinity than in functional yield (Fig. 4). The yields of variants GRIA ($B_{max}$ of 1446 ± 78) and SLNIG ($B_{max}$ of 1372 ± 34) are 1.2-fold greater than the wild-type A$_2$aR yield ($B_{max}$ of 1186 ± 103) (Fig. 4). Similarly, variants Q89A$^{3.37}$ and Q89S$^{3.37}$ exhibit $B_{max}$ values (1677 ± 37 and 2018 ± 82, respectively) that are 1.4–1.7-fold greater than that of the wild-type receptor. Accordingly, we posit that improvements in ligand-binding affinity, compared with functional yield, influenced variant enrichment to a greater extent in our screen. While variants GRIA, SLNIG, and Q89S$^{3.37}$ bind $^3$H-NECA with similar affinities, GRIA and SLNIG are present in the PS4 population with ~17-fold and 12-fold lower abundance, respectively. Thus, our identification of the variants GRIA and SLNIG underscores the sensitivity of our screen and the utility of our analytical method to directly account for epistasis in a search for protein fitness optima.

**Epistasis restricts evolutionary paths to enriched variants**. Finally, we sought to identify evolutionary trajectories leading to enriched variants and the extent of epistasis within this complex network. Accordingly, we generated a network diagram comprising variants in PS4 with enrichment rates ≥1 (Fig. 5a). In this diagram, each node represents a unique variant, the node's radius scales with the variant's enrichment rate, and each edge represents a difference in one amino acid (i.e., one Hamming distance). The color of each edge is that of the variant with lower enrichment, portraying the direction that would be favored in Darwinian evolution, or from less fit to more fit. Although each of the most highly enriched variants is accessible by a single mutational step from the lowest network level (Fig. 5a), these variants are rare and often accessible by few trajectories as expected[5]. While functional variants absent in our processed data set may exhibit significantly improved function or provide additional paths to enriched variants, we find that variants with modest enrichment

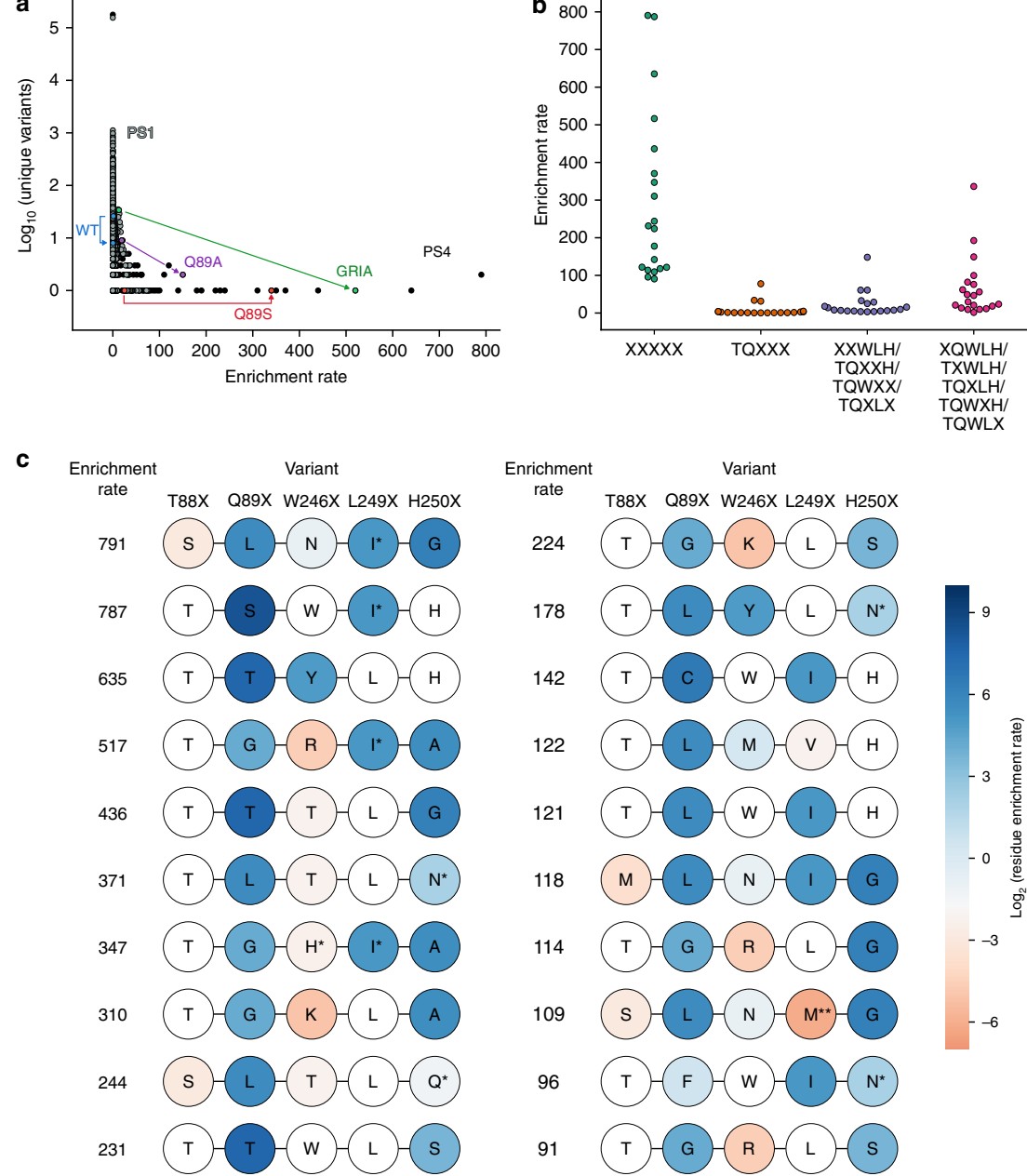

**Fig. 3 BRIDGE reveals highly enriched A₂aR variants due to epistasis between distal sites. a** A histogram of the $\log_{10}$(count) of unique variants sharing a given enrichment rate demonstrates stringent, progressive enrichment of few highly enriched mutants from the naive library to the post sort (PS) 1 (gray) and PS4 (black) libraries. A small number of variants are highly enriched in PS4 (e.g., GRIA), while variants with wild-type (WT)-like behavior are not expected to be highly enriched. A positive control (Q89A) and the most abundant variant in PS4 (Q89S) also demonstrate relatively high enrichment as expected. **b** The top 20 variants containing at least two distal, mutated residues (XXXXX) exhibit far greater enrichment rates compared to the top 20 variants containing proximal triple (TQXXX), proximal double (e.g., XXWLH), or single (e.g., XQWLH) substitutions. **c** Substitutions that are deleterious in isolation are common in the 20 most highly enriched A₂aR variants containing mutations in distal sites. Residues that are beneficial or deleterious as single substitutions are colored according to their $\log_2$-fold enrichment, where the color spectrum is centered around the wild-type enrichment rate. For example, variant SLNIG is enriched 791-fold, but the single T88S (SQWLH) substitution is deleterious. Wild-type residues have a white background. *Residue abundance between 7 and 11 in the naive library, **not detected in PS4.

rates are highly interconnected (Supplementary Fig. 12). Thus, the scarcity of edges and disconnected networks associated with the highly enriched variants (Fig. 5b) suggest significant contribution of epistasis to variant fitness. It is only specific combinations of residues that result in such pronounced enrichment rates, while in most other contexts (i.e., sequence backgrounds) an individual residue is either insufficiently beneficial or even deleterious to the variant's enrichment. In order to better illustrate this point, we generated the isolated network of variant TSWLH (Fig. 5c), which is accessed through several paths originating from a highly interconnected network of functional parents (Fig. 5c; Supplementary Fig. 12). Projecting this network onto a three-dimensional fitness landscape would result in a broad, "Fujiyama"-like terrain[1], where several parents benefit

comparably from an X88T[3.36] or X89S[3.37] mutation and, to a lesser extent, an X246W[6.48] or X249L[6.51] substitution.

## Discussion

In this work, we present a DMS methodology that overcomes NGS read-length limitations to match distal mutations from the same variant without modification of standard experimental protocols. Following standard NGS library preparation, BRIDGE employs an in silico pipeline to match nonoverlapping paired-end reads by their sequencing chip surface coordinates, which will be

| Variant | Enrichment rate | Fold increase in enrichment rate over WT |
|---|---|---|
| SLNIG | 791 | 2635 |
| TSWIH | 787 | 2624 |
| TTYLH | 635 | 2117 |
| TGRIA | 517 | 1722 |
| TTTLG | 436 | 1454 |
| TLTLN | 371 | 1235 |
| TGHIA | 347 | 1157 |
| TSWLH[a] | 336 | 1121 |
| TGKLA | 310 | 1034 |
| SLTLQ | 244 | 813 |
| TTWLS | 231 | 772 |
| TGKLS | 224 | 747 |
| TTWLH[a] | 193 | 642 |
| TLYLN | 178 | 592 |
| TAWLH[a] | 149 | 497 |
| TQYLG[b] | 148 | 494 |
| TCWIH | 142 | 474 |
| TLMVH | 122 | 406 |
| TLWIH | 121 | 405 |
| MLNIG | 118 | 393 |
| TGRLG | 114 | 379 |
| SLNMG | 109 | 365 |
| TCWLH[a] | 100 | 332 |
| TFWIN | 96 | 319 |
| TGRLS | 91 | 302 |

Table 1 Top 25 variants ranked by enrichment rate from the naive to the PS4 library.

[a]Single-residue mutant, [b]Proximal double-residue mutant.

an exact match due to the proximity imposed by the bridge amplification mechanism. The physical location of each read is encoded in FASTQ files generated by Illumina sequencers; thus, this method can be applied to any Illumina SBS experiment. Since single reads often produce greater sequencing error rates, we designed NGS primers to group mutated bases near the 5′-end of each read, where error rates are low. Using this strategy, we reduce local error rates to values comparable with those sought using overlapping reads.

To demonstrate its utility, we use BRIDGE to mutate five distal residues within a GPCR ligand-binding pocket yielding variants with enrichment rates up to 2600-fold greater than that of the wild-type receptor. Characterization of highly enriched variants using radioligand binding confirmed fourfold decreases in $K_d$ to a target ligand and validated our methodology for engineering improved binding affinity in a human GPCR. Importantly, combinations of mutations in distal sites confer the greatest improvements in enrichment, and the majority (13/20) of the most highly enriched variants with distal mutations contain a substitution that is deleterious alone. As a result, these 13 variants would not necessarily be generated using the conventional approach of combining individual gain-of-function substitutions over the course of several rounds of directed evolution. However, these interactions are often a necessary step along a complex evolutionary path that is not accessible or obvious by single-site, step-wise mutagenesis[4,5]. Thus, the capacity to directly identify functional variants with unobvious combinations of residues is critical, considering it is exceedingly difficult to predict high-order (i.e., more than two residues) epistasis through computational methods[3,50].

This contribution represents the fourth study of a GPCR using DMS[47,51,52], but is the first to use DMS to investigate epistasis, as previous efforts were limited to single-site-saturation mutagenesis libraries. Although all five mutated residues are highly conserved among $A_2aR$ orthologs, the number and diversity of tolerated substitutions varies considerably at each site. For example, variants with enrichment rates exceeding that of wild-type exhibit a strong preference at position T88[3.36] for threonine or serine, which contain side chains dominated by a hydroxyl group. In contrast, these variants sample glycine, histidine, serine, alanine, and proline with similar frequencies at position H250[6.52]. Strikingly, one of the most permissive sites among enriched variants is W246[6.48], which is part of the highly conserved CWxP motif in

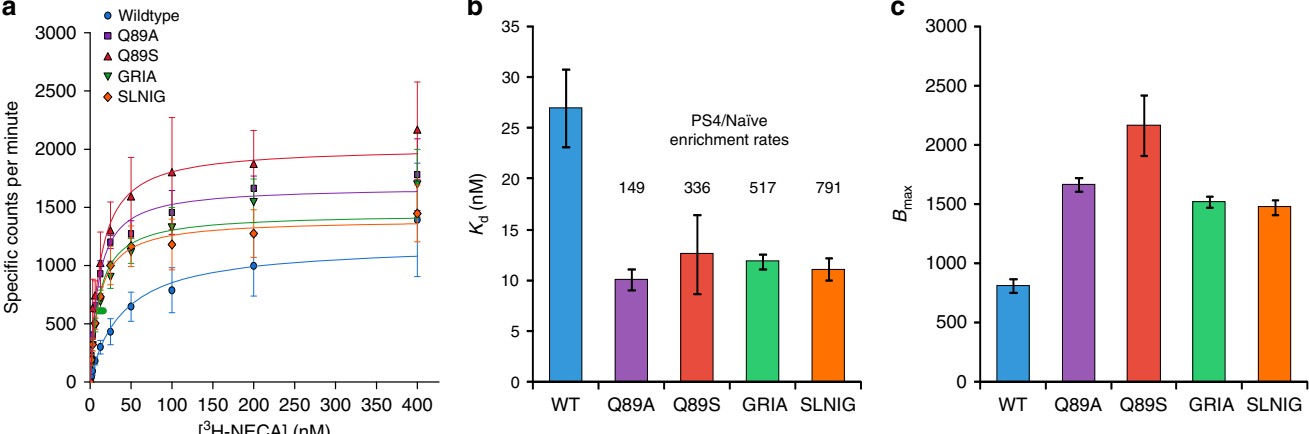

**Fig. 4 Radioligand binding confirms improved ligand-binding affinities. a, b** Compared with wild-type $A_2aR$, highly enriched variants, Q89A/S, GRIA, and SLNIG bind [3H]-NECA with fourfold greater affinity ($K_d$). **c** Functional yields ($B_{max}$) of the highly enriched variants are 1.2–1.7-fold of the wild-type receptor yield, suggesting improvements in ligand-binding affinity rather than functional yield dictate variant enrichment. Data represent the mean of three biological replicates, and error bars represent their standard deviation. Source Data are provided as a Source Data file.

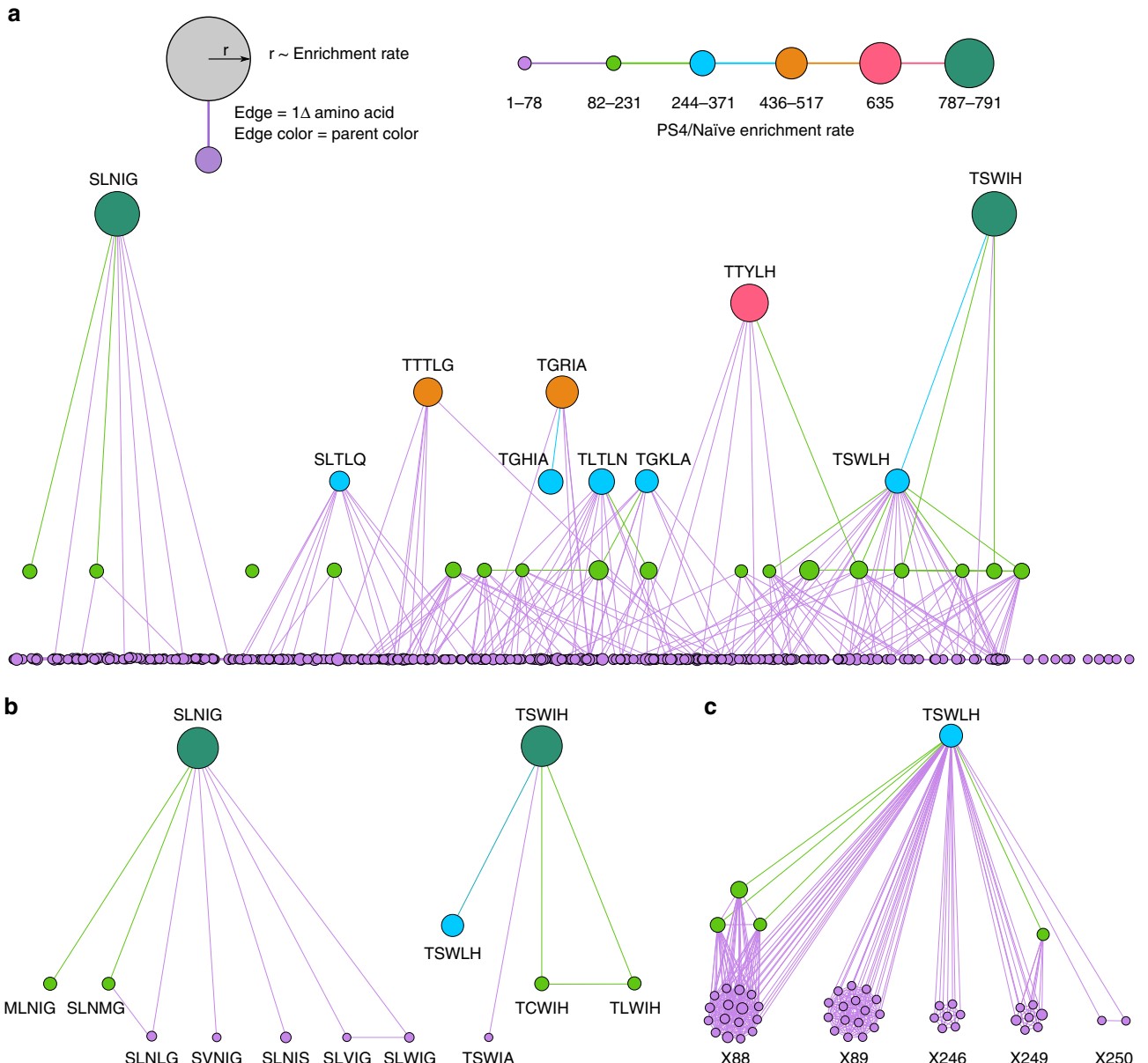

**Fig. 5 Network analysis of enriched variants reveals rare paths to highly enriched variants. a** A network diagram comprising Post Sort (PS) 4 variants with enrichment rate ≥1. Each node corresponds to a unique variant, and the node's radius scales with the variant's enrichment rate. Each edge represents a difference in one amino acid, and the color of each edge is the same as the variant with lower enrichment, providing a sense of the trajectory taken by Darwinian evolution, or from less fit to more fit. As anticipated, variants with the greatest fitness are rare and often accessible by only few paths (edges), reflecting a rugged sequence–function landscape indicative of epistasis. **b** Isolated networks of variants SLNIG and TSWIH underscore the scarcity of paths originating from disconnected functional variants, which suggests that specific combinations of residues lead to pronounced improvements in function rather than a few critical residues. This observation is indicative of epistasis, where the context (i.e., background sequence) of a protein dictates an individual residue's effect on the protein's phenotype. **c** In contrast, TSWLH is accessible by numerous paths reflecting a broad sequence–function landscape. Parent variants leading to TSWLH are highly interconnected as several related proteins benefit similarly from the same substitution.

GPCRs[44]. This conserved tryptophan residue in TM6 has been hypothesized to act as a "rotamer toggle switch"[45] or, alternatively, part of a "transmission switch"[46] in class A GPCRs, mediating receptor activation through modulation of a proline-induced kink or horizontal rotation of TM6, respectively. The existence of a rotamer toggle switch has been the subject of debate with a toggle switch mechanism observed in simulations[53] and an alternative motion inferred from an agonist-bound $A_2aR$ structure[43]. Our data set can be used to inform or support hypotheses generated from experimental and computational efforts addressing this important, putative mechanism. For example, by

accepting the rotamer toggle switch model, enrichment of the single W246Y substitution may reflect preserved functionality of this mechanism, and is in agreement with a previous study, wherein potency of the agonist CGS21680 was comparable to that of the wild-type receptor[54]. In contrast, nonaromatic substitutions at W246 may adopt active conformations that are stabilized by particular covaried residues due to epistasis.

The $W246^{6.48}$ residue in $A_2aR$ is also thought to play a role in the allosteric modulation of the receptor through interaction with ligands in an allosteric binding pocket[55]. Binding of $Na^+$ or amiloride appears to impede agonist binding by stabilizing an

inactive conformation[55,56]. Moreover, molecular dynamics simulations suggest W246[6.48] is the only residue that simultaneously interacts with amiloride and an inverse agonist, ZM241385, in the orthosteric pocket, underscoring its role in mediating receptor allostery[56]. We demonstrate the capacity to mutate W246[6.48] while improving agonist binding affinity, which represents a potential approach to engineering allostery, a therapeutically relevant mechanism[57]. Indeed, a W246A[6.48] substitution enabled concurrent binding of $Na^+$ and $^3H$-NECA and improved ZM241385 binding in the presence of $Na^+$[58]. Concurrent mutagenesis of residues in the orthosteric and allosteric binding pockets, which span several transmembrane helices, could enable fine-tuning of the allosteric response to molecules ideal for therapy (e.g., orthogonal to native physiology).

Our findings support a view of conformational plasticity[59] and tolerance to substitutions in evolutionarily conserved sites[47]. This plasticity is thought to be a result of the evolutionary constraints imposed on GPCRs, which sample a complex conformational ensemble to mediate finely tuned functions. In our screen, we relax several constraints necessitating only proper folding, trafficking to the plasma membrane, and agonist binding. Thus, while increased tolerance to substitutions may be expected, we observe a small number (740) of variants that exhibit improved agonist binding reflecting the inherent constraints imposed by epistasis that are not revealed through single-site mutagenesis. The selective pressures imposed in our screen may also point to putative functional effects of epistasis on variant phenotypes and enrichment rates. In different sequence backgrounds, residues contributing to enrichment may instead promote impaired folding, thermodynamic stability, trafficking, or capacity to bind the fluorescent ligand. The scarcity of permissible sequences can be attributed to the rarity of compatible combinations of functionally beneficial, but often destabilizing, mutations and compensatory, stabilizing mutations[60]. For example, the evolution of cortisol specificity in the glucocorticoid receptor was mediated by a rare combination of seven mutations[61,62]. Among these seven mutations, five are responsible for altered specificity but render the receptor non-functional in the absence of the two remaining stabilizing mutations, which are highly specific due, at least in part, to their recruitment of structural rearrangements that directly compensate for conformational changes induced by the destabilizing mutations. The absence of the stabilizing mutations did not significantly affect receptor expression levels suggesting the conformational change, rather than reduced protein degradation or aggregation, permitted functional cortisol specificity[62]. The locations of the mutated sites in the $A_{2a}R$-binding pocket may also facilitate specific epistatic interactions[8]. In other words, interactions between particular substitutions and the ligand or nearby residues may yield specific conformational changes leading to improved function. However, permissible substitutions are only tolerated in the presence of specific residues that act to stabilize the protein variant or properly coordinate atomic interactions with the ligand. Notably, this mechanism is not mutually exclusive with alternative mechanisms underlying epistasis such as rescuing proper surface expression as observed in the neuraminidase protein in oseltamivir-resistant strains of H1N1 influenza[63]. Indeed, our observations may also be affected by nonspecific epistatic interactions[8], which contribute to global effects on protein biophysical properties and are independent of the site of mutation. To conclusively determine the type and strength of epistatic interactions occurring in each variant, additional experiments must be conducted. However, toward elucidating the structure–function relationship of GPCRs, our approach represents a needed complement to the rapidly increasing number of GPCR crystal structures, which are often produced following extensive modification of the native

proteins[64]. BRIDGE will be of particular importance in rapidly constructing sequence–function landscapes for important domains in GPCRs that are reticent to crystallization, which represent the vast majority of this protein superfamily.

Although long-read technologies (e.g., PacBio and nanopore sequencing) are available, they are currently better served for genome assembly than the detection and quantification of rare variants due to their low-throughput and sequencing accuracy[65,66]. Previously established methods that enable construction of full-length genes from short NGS reads require highly specialized instruments and expertise or work-intensive manipulation of libraries, which risks errored analyses due to unsuccessful or incomplete manipulation. As a result, although these approaches represent clever solutions to a difficult problem, their disadvantages have limited their application. BRIDGE provides an accessible approach to important problems requiring sequence information for distal mutated positions such as investigation of catalytic and binding sites or covariation in allosteric and orthosteric sites. Currently, BRIDGE cannot be used to sequence the full-length of genes >600 bp. We also note practical considerations in applying BRIDGE to protein libraries (Supplementary Note 3). However, this method is immediately useful for mutagenesis at several distal sites including site saturation, which is often limited to few positions by transformation efficiency and screening capacity. Nonetheless, as advancements in sequencing technologies increase read lengths and improve accuracy, we expect that our approach will scale to enable investigation of high-order epistasis of full-length genes and pathways at low cost and high fidelity.

## Methods

**Plasmid construction.** In order to construct pYC $A_{2a}R$, the prepro-$A_{2a}R$-His$_{10}$ coding sequence was amplified from pITy $A_{2a}R$[67] using primers 1 and 2, which introduce flanking HindIII and XhoI sites, respectively. The amplicon and the pYC2/CT vector backbone (Invitrogen) were digested, ligated, and transformed into E. coli DH5α using standard molecular cloning techniques. All pYC $A_{2a}R$ variants described in this study were constructed using PCR-driven overlap extension[68]. Briefly, mutations were introduced into $A_{2a}R$ by amplifying prepro-$A_{2a}R$-His$_{10}$ from pYC $A_{2a}R$ using a subset of primers 1–18 (Supplementary Table 6). The full-length prepro-$A_{2a}R$-His$_{10}$ amplicons contain flanking HindIII and XhoI sites, which were used to clone the DNA into pYC2/CT.

**Yeast strains and culturing conditions.** Saccharomyces cerevisiae strain BJ5465 (MATa ura3-52 trp1 leu2Δ1 hisΔ200 pep4::HIS3 prbΔ1.6 R can1) (ATCC) was used for all experiments conducted in this study. Strains harboring pITy $A_{2a}R$ were maintained in YPD medium. Strains harboring pYC $A_{2a}R$ were maintained in synthetic dextrose medium supplemented with casamino acids lacking uracil (SD-ura)[69]. In order to induce gene expression, cultures were initially incubated overnight at 30 °C with shaking. Following overnight incubation, strains were subcultured into 5 mL of YPR or SR-ura containing 2% (w/v) raffinose at an initial $OD_{600}$ of 0.5. Once culture $OD_{600}$ reached 2–3, typically 10–12 h, gene expression was induced by subculturing strains into 5 mL of YPRG or SRG-ura containing 2% (w/v) raffinose and 2% (w/v) galactose at an initial $OD_{600}$ of 0.2. Gene expression was carried out for 12 h.

**Fluorescent ligand binding.** Yeast cells producing $A_{2a}R$ were diluted to an $OD_{600}$ of 1.0 in 1 mL binding buffer (50 mM Tris-HCl, 10 mM $MgCl_2$, pH = 7.5) and incubated with FITC-APEC[22] (NIMH, Bethesda, MD) with gentle mixing while protected from light at room temperature for 1 h. Cells were then placed on ice for 30 min prior to resuspension in 1 mL ice-cold 1× phosphate buffered saline (PBS, pH = 7.5) and immediate analysis using FACS. All FACS analyses were performed using a BD FACSAria I flow cytometer using a 488 nm laser and 530/30 nm bandpass filter.

**Pilot enrichment experiment.** Yeast cells harboring wild-type $A_{2a}R$ and an inactive variant (pITy $A_{2a}R$ C28A/C82A/C128A/C185A/C245S/C254A/C394S) in the pITy backbone were induced for gene expression, as described above. The $OD_{600}$ of each culture was measured and used to estimate cell density. Cells were resuspended in binding buffer (50 mM Tris-HCl, 10 mM $MgCl_2$, pH = 7.5) and combined to generate mixtures of cells producing wild-type and inactive $A_{2a}R$ at ratios of 0:1, 1:0, 1:10, 1:100, and 1:1000. Each mixture had a final $OD_{600}$ of 1 in 1 mL binding buffer. Cells were incubated with 10 µM FITC-APEC as described

above and sorted using FACS. The pure (0:1 and 1:0) samples of cells producing inactive and WT A$_2$aR were used to set the sort gate such that it enclosed ~0.5% of cells producing the inactive variant. Approximately 10,000 cells were sorted into 5 mL SD-ura supplemented with ampicillin (100 µg/mL). Cells were incubated at 30 °C to repeat the culturing and FACS steps.

**Library construction**. The A$_2$aR library was constructed using primers 1 and 2 and degenerate NNS primers 19–22 using PCR-driven overlap extension[68] resulting in site-saturation mutagenesis of five residues (T88, Q89, W246, L249, and H250). After PCR, the amplicon was purified and concentrated using a Zymo DNA Clean and Concentrator kit, digested with *HindIII* and *XhoI*, and purified by gel extraction. Approximately 12 µg of DNA was ligated into 7.5 µg of *HindIII*- and *XhoI*-digested pYC2/CT and electroporated into *E. coli* DH5α. The library DNA was extracted from *E. coli* using a Zyppy Plasmid Maxiprep Kit, and ~120 µg of DNA was transformed into yeast cells using the high-efficiency lithium acetate protocol[70]. Transformed cells were used to inoculate 500 mL SD-ura supplemented with 100 µg/mL ampicillin. After incubation at 30 °C for 24 h, ~10$^9$ cells from the naive library were used to inoculate 50 mL SD-ura with 100 µg/mL ampicillin for FACS screening.

**Library screening and fluorescence-activated cell sorting**. Yeast harboring the pYC A$_2$aR library were induced for gene expression in 50 mL cultures, and ~10$^8$ cells were incubated with fluorescent ligand using protocols described above. For each round of screening, yeast harboring empty pYC vector and wild-type pYC A$_2$aR were induced for gene expression and incubated with 10 µM FITC-APEC as described above. Yeast cells were resuspended in 1× PBS (pH = 7.5) at an OD$_{600}$ of 1.0 and screened using FACS. In the first round of screening, ~50,000 cells were collected, while ~100,000 cells were collected in each subsequent round of screening. Cells were collected in SD-ura supplemented with ampicillin (100 µg/mL) in order to repress P$_{GAL1}$-driven gene expression mitigating artifacts due to varying degrees of metabolic burden imparted by A$_2$aR variants. During the first two rounds of sorting, yeast harboring the A$_2$aR library were sorted to collect all cells with green fluorescence intensities >~99.5% of the negative control. In the third round of screening, the post sort (PS) 2 population was sorted to collect all cells with fluorescence intensities above background (low-stringency) or 5% of the population with the greatest fluorescence intensities (high stringency). In the fourth round of screening, the PS3 populations isolated using low- or high-stringency gating were sorted using the same strategy as in the previous round.

**Deep sequencing of library**. Plasmids were extracted from the naïve and sorted populations using a Zymo Yeast Plasmid Miniprep II kit according to the manufacturer's instructions. Plasmid yield was estimated through transformation of extracted plasmid into *E. coli* DH5α, determination of colony-forming units (CFUs), and comparison with CFUs obtained from transformation of *E. coli* using 50 ng pYC A$_2$aR. Extracted DNA was digested with *EcoRI* and *HindIII* to fragment contaminating gDNA and improve amplification of the target region (Supplementary Fig. 13). The amplicon library was prepared for sequencing using the 16S Metagenomic Sequencing Library Preparation guide (Illumina). First, amplicons were generated using primers 23–30 to append Illumina adapter sequences with the following protocol: 98 °C for 3 min, followed by 20 cycles of 98 °C for 10 s, 68.5 °C for 30 s, and 72 °C for 30 s, followed by 72 °C for 5 min. In all, 0–3 nucleotides were added after the adapter in each primer to facilitate cluster identification during sequencing. Approximately 22 ng (ca.) template was used for the naive library, while approximately 0.78 ng (ca.) template was used for each sorted population. After verifying the specificity of the PCR using an Agilent Tapestation, the PCR product was purified using AMPure XP beads according to the manufacturer's instructions. Subsequently, dual indices and sequencing adapters were appended to the purified amplicons using a Nextera XT Index kit with 5 µL of the resuspended amplicon as a template. Phusion polymerase was used for the PCR with the following protocol: 98 °C for 3 min, followed by 8 cycles of 98 °C for 30 s, 55 °C for 30 s, and 72 °C for 30 s, followed by 72 °C for 5 min. PCR products were purified using AMPure XP beads and their expected sizes verified using an Agilent Tapestation. DNA was quantified using a Qubit fluorometer and normalized such that the naïve library comprised 80% of the total DNA, while the sorted populations evenly comprised the remaining 20%. The combined library was sequenced using a NextSeq 500 and the v2 high output kit for 150 cycles of 2 × 75 bp sequencing.

**NGS data analysis**. Custom python code was used to process raw sequencing data. First, we leverage the use of 5′-focused mutations by trimming forward and reverse reads to retain sequences near the 5′-end of each read. As a result, we retain reads that may otherwise be discarded in downstream quality-filtering steps due to low Q scores near the 3′-end. Trimmed forward reads contain "CGTCCTGG" and "CCATCTTCA" at their 5′- and 3′-ends, respectively. Trimmed reverse reads contain "GCAGTTGA" and "GCAGAGG" at their 5′- and 3′-ends, respectively. Trimmed sequences containing insertions or deletions were discarded. The remaining sequences were quality-filtered using a minimum quality score of 20. Reverse reads were reverse-complemented and paired to corresponding forward reads by matching the (*x*, *y*) coordinate values in the FASTQ sequence identifiers. The processed data for post sort populations 1–4 was analyzed using

Enrich2 software with the weighted least-squares scoring method, wild-type normalization method, and a minimum read count of 15. Due to the size of our library, a significant number of loss-of-function mutations are encoded in the naive library; thus, the enrichment rates of functional mutants in the first round of screening will be disproportionately high compared with those in the remaining rounds. As Enrich2 rank-orders variants based on linear regression of log-transformed variant counts, we performed the analysis excluding sequencing data for the naive library to avoid introducing nonlinearity into the fit data. To minimize noise due to low counts, a minimum of 15 reads in the naive library was used as a threshold for calculating variant enrichment rates. Estimation of the non-specific carryover rate ($\lambda$) was carried out as described by Jolma et al.[71]. Specifically, $\lambda$ was calculated using the equation $f_{k+1} = \lambda f_k$, where $f_{k+1}$ and $f_k$ represent the fraction of reads that contain at least one stop codon in a given round and the preceding round, respectively. Codon adaptation index was calculated using open source code[72] against the reference set of conserved yeast genes provided by Sharp et al.[41]. Average error rates and quality scores were determined by initially calculating the arithmetic mean of the error rate for each region of interest. Then, this value was converted to an average quality score. To evaluate the sequencing accuracy for constant bases flanking mutated bases, the sequences "TCCTC" and "AGCT" were used for the forward read, and "TGAT" and "GGGCAG" for the reverse read. Gephi 0.9.2 was used to generate the variant network figures and the force-directed graph. Figure 3a was generated by first running the force atlas algorithm to cluster interconnected nodes, then the Network Splitter 3D algorithm was used to separate variants by enrichment rate into network levels. Figure 3b was generated by masking all nodes, except for SLNIG or TSWIH as well as variants separated by 1 hamming distance from either variant. Figure 3c was generated by first isolating variant TSWLH and variants with lower enrichment rates separated by 1 hamming distance. The force atlas algorithm was used to cluster nodes in the same network level identified previously. The force atlas algorithm was used to cluster highly interconnected nodes.

**Sequence alignment and logo plots**. Amino acid sequence alignments were generated using GPCRdb and jalview. Frequency logo plots were generated using WebLogo Version 2.8.2.

**Radioligand binding**. Whole-cell radioligand binding was performed as described by Niebauer et al.[73] with few modifications. Briefly, gene expression was induced in cells harboring pYC or pYC A$_2$aR variants using the protocol described above. Cells were then washed with 1X PBS (pH 5.8) and resuspended in ligand-binding buffer (50 mM MES-Tris, 10 mM MgCl$_2$, pH 5.8) at a final concentration of 1 OD. Cells were incubated with 0–400 nM [$^3$H]-NECA (Perkin Elmer) for 1.5 h at room temperature. Opti-Fluor scintillation cocktail (Perkin Elmer) was added to each well, and radioactivity was measured as counts per minute using a PE-Wallac Microbeta Trilux 1450-024 microplate scintillation counter. Specific signal was calculated by subtracting counts for the empty vector negative control from those for each variant at each ligand concentration. Data were fit to a one-site binding model using instrumental weighting using Origin 9.

**Reporting summary**. Further information on research design is available in the Nature Research Reporting Summary linked to this article.

## Data availability

Raw sequencing data generated in this study have been deposited in the NCBI Sequence Read Archive under accession number PRJNA545342. All other relevant data are available from the authors upon reasonable request.

## Code availability

Python code generated in this work is available from https://github.com/j-yoo/BRIDGE.

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

## Acknowledgements

The authors acknowledge funding support from the National Institute of General Medical Sciences of the National Institutes of Health under Award Number R01GM116128-01. J.I.Y. acknowledges support from a National Science Foundation Graduate Research Fellowship under grant No. 1650114. The authors are grateful for the assistance of Dr. Jennifer Smith, manager of the Biological Nanostructures Laboratory within the California NanoSystems Institute, supported by the University of California, Santa Barbara and the University of California, Office of the President. The fluorescent ligand FITC-APEC was generously provided by the NIMH Chemical Synthesis and Drug Supply Program. We thank Dr. Sean Gilmore for proposing the proximity-based matching of paired-end sequencing reads and St. Elmo Wilken for discussions on coding. We are grateful to Professor Kevin Plaxco and Professor Arnab Mukherjee for critical feedback on figures and manuscript drafts.

## Author contributions

J.I.Y., P.S.D., and M.A.O. conceived the project, designed experiments, analyzed the data, and contributed to writing and reviewing the paper. J.I.Y. performed all experiments and wrote the code described in the paper.

## Competing interests

The authors declare no competing interests.
