## [Peer Review File · Nature Communications]

Reviewers' Comments:

Reviewer #1:

Remarks to the Author:

The authors explore the question of epistasis, using ligand binding of the A2A GPCR as an example of function that could potentially be solved by another combination of the amino acids in the ligand binding pocket.

Using available crystal structure of A2A in complex with an agonist NECA complex, they identified five amino acid positions that contributed key interactions with the ligand. Subsequently, they created a saturation mutagenesis library of various combinations of all possible substitutions in these positions. Using a fluorescent ligand binding and cell sorting, they selected clones that were overexpressed and/or bound the ligand better. Using NGS, they identified a number of novel variants of the receptor that retain the ability to bind ligand, and are enriched. For this, they developed a novel method BRIDGE to link mutated positions on the same DNA molecule that are separated by longer than the sequencing read of the Illumina system.

They then characterised the ligand binding of selected 3 variants (one with 4 out of 5 substitutions, denoted GLIA), and two single point mutants in the second position.

Based on the fact that they identified a high affinity binding variant GLIA, they have shown a case of epistasis (getting a similar or better functionality by finding another unique combination of amino acids, and that this functionality could not be achieved by sequential addition of mutations).

This is an interesting problem, and certainly the engineered variants of the receptor may find some useful applications in the future. Having said this, not all the supporting data are shown or are of sufficient quality to fully justify the conclusions in the current version of the manuscript. The following suggestions would help to improve it:

The biological function of the agonist binding is also to activate the receptor. It would be interesting to see if the evolved variants can activate G proteins.

At 10 μ M ligand, all possible sites (at least for WT or better binding variants) should be saturated so the selection was mainly done on expression levels. This point is not accurately discussed in the paper.

A more detailed explanation of the BRIDGE method (for the non-expert in NGS technologies) would help.

Either by chance or design, the five mutagenesis positions were grouped (XX- 750 bp gap-XXX), therefore creating the simplest case scenario for the method. A brief discussion of how it can be extended to the case of three separate groups of mutations would be helpful.

Figure 3c and Table 1 seem to duplicate each other.

Figure 4 shows a comparable level of expression as measured by radioligand binding (B_{max}) (ca 70% of the WT) but in the text the authors state 17-fold decrease. It is not clear which data support this statement.

Why was the most enriched variant SLNIG not included in the ligand binding assays?

There is a statement in the text that individual mutations were all detrimental but no data to support this are shown.

The quality of the ligand binding data shown does not allow the accurate determination of the K_d s,

especially for the key variant GLIA, on which a lot of subsequent analysis and claims are based. The data clearly suggest that it binds but the quality needs to be improved. As a suggestion, homogeneous fluorescence based binding assays (HTRF and nanoBRET) are available for A2A and are probably superior to the radioligand binding in throughput. Additionally, it is not clear from the legend if the data shown represent total binding or if they were corrected for the non-specific binding (although it says so in the methods).

In the discussion, there is a statement that the enrichment for some sequences was up to 2,600-fold. However, no such data is shown.

Reviewer #2:

Remarks to the Author:

Naturally evolved proteins are typically metastable, and most random mutations destabilize native protein structures. For these reasons, evolutionary pathways are highly constrained, and many sequences with useful activities are inaccessible. Next generation sequencing tools have given rise to new experimental approaches including deep mutational scanning (DMS), which are providing unprecedented insights into the mechanisms of protein evolution. DMS technologies have also opened new doors for protein design. In this work, Yoo et al. develop a novel sequencing technique that facilitates facile DMS of sequence-distant sites using short, paired-end reads on conventional Illumina sequencing platforms. They then apply this approach to evaluate the effects of higher order epistatic interactions on the binding affinity of the adenosine A2a receptor, which is an important G-protein coupled receptor. Their results identify new mutant receptors that significantly increase the binding affinity of the receptor for a specific ligand, and illuminate the evolutionary barriers to the sampling these specific variants. This work is generally of very high quality and originality, and offers both methodological advances as well as some fundamental insights. I expect it to have broad relevance to several biochemical and biophysical research communities. Nevertheless, I have several suggestions to improve the manuscript:

-The higher order epistasis illuminated herein is interesting, and I think the authors do a particularly good job of explaining these connections graphically. However, the authors do not address the potential epistatic mechanisms at all. Elucidation of epistatic mechanism is perhaps beyond the scope of this investigation. Nevertheless, the authors could at least discuss the potential mechanisms (ie folding versus binding). Based on the collective work of the Tawfik, Arnold, and Bloom groups, we may expect most of these epistatic effects to arise from folding constraints. If this were the case, one might expect most of these single variants that are poorly-tolerated in isolation to compromise expression. This possibility could be easily tested by measuring the B_{max} values for a few of the single variants that are depleted from the pool. A comparison of a handful of select Q89 versus W246 mutations could help to illuminate the reasons why most of these pathways seem to be forbidden- do they express but not bind, or do most of these individual substitutions compromise expression, or both? This exercise would significantly enhance the relevance of these results considering the scarcity of deep mutational scanning data for integral membrane proteins.

-The sequencing platform described herein is unlikely to be broadly impactful in the long-run given the impending rise of sequencing technologies with longer reads (ie Oxford Nanopore and PacBio). Nevertheless, the author has correctly pointed out that these approaches are somewhat boutique and expensive at the moment. For this reason, the BRIDGE technique described herein will likely prove useful for certain experiments. However, the reader would benefit from some discussion on the practical limitations of this approach. Are there likely to be size/ distance restraints for matching the paired end reads? Can you pair reads that are 500 bp apart (likely), 1 kb, 10 kb? How large can the amplicons get before it becomes difficult to match, or does this matter at all? Does the density of clusters on the cell impact your ability to match reads? These may be

considerations that are obvious to the authors, but they may not be so obvious to the casual reader.

-The authors describe the structural context of the chosen side chains, but it is difficult to visualize without a reference structure. I believe there is at least one nice structure for this receptor, and if not, it should be easy enough to make a homology model of this specific receptor using iTASSER. Either would be useful as a guide for the more structurally-minded readers, and the authors should include a structural figure for reference.

Overall, I enjoyed reading this paper. It is very well written, and most importantly, is of excellent quality. Both the methodological advances and basic discoveries are likely to be broadly relevant to those interested in protein evolution and design.

Reviewers' comments:

Reviewer #1 (Remarks to the Author):

The authors explore the question of epistasis, using ligand binding of the A_{2A} GPCR as a example of function that could potentially be solved by another combination of the amino acids in the ligand binding pocket.

Using available crystal structure of A_{2A} in complex with an agonist NECA complex, they identified five amino acid positions that contributed key interactions with the ligand. Subsequently, they created a saturation mutagenesis library of various combination of all possible substitutions in these positions. Using a fluorescent ligand binding and cell sorting, they selected clones that were overexpression and/or binding the ligand better. Using NGS, they identified a number of novel variants of the receptor that retain the ability to bind ligand, and are enriched. For this, they developed a novel method BRIDGE to link mutated positions on the same DNA molecule that are separated by longer than the sequencing read of the Illumina system.

They that characterised the ligand binding of selected 3 variants (one with 4 out of 5 substitutions, denoted GLIA), and two single point mutants in the second position.

Based on the fact that they identified a high affinity binding variant GLIA, they have shown a case of epistasis (getting a similar or better functionality by finding another unique combination of amino acids, and that this functionality could not be achieved by sequential addition of mutations).

This is an interesting problem, and certainly the engineered variants of the receptor may find some useful applications in the future. Having said this, not all the supporting data are shown or are of sufficient quality to fully justify the conclusions in the current version of the manuscript.

The following suggestions would help to improve it:

We thank the reviewer for their careful consideration of our manuscript and for their helpful comments and suggestions. Below, we address the reviewer's comments point-by-point.

The biological function of the agonist binding is also to activate the receptor. It would be interesting to see if the evolved variants can activate G proteins.

This is an important consideration, and we agree that determination of faithful G protein activation is of interest for both fundamental and applied studies. In order to examine the functional G protein coupling of the enriched, engineered adenosine A_{2a} receptor (A_{2a}R) variants, we attempted additional experiments; however, these experiments were hindered by poor transformation of constructs into engineered yeast strains as well as unavoidable aggregation phenotypes of the transformed yeast strains that prohibited accurate ligand binding experiments.

In this experiment, we sought to examine the capacity to activate G protein signaling in wildtype A_{2a}R as well as the variants characterized using radioligand binding (i.e. Q89A, Q89S, GRIA, and SLNIG, which was added in new radioligand binding experiments). We planned to carry out the G protein signaling experiment in *Saccharomyces cerevisiae*, which offers a native GPCR signaling pathway, the pheromone response, commonly used to assay the function of mammalian GPCRs. To facilitate this experiment, we designed and constructed CRISPR/Cas9 plasmids to engineer the native pheromone signaling pathway, which enhances coupling to A_{2a}R. Specifically, we knocked out the native mating receptor, *Ste2*, along

with genes responsible for $G\alpha$ deactivation, *Sst2*, and cell cycle arrest, *Far1*. These genetic modifications are commonplace in coupling heterologous GPCRs to the native yeast pheromone pathway.

Upon repeated transformation of the engineered yeast strains with A_2aR variants and a GFP reporter construct (P_{FUS1} -eGFP), however, we were unable to obtain transformants. We believe this issue may be due to the reporter construct; however, the engineered yeast strain is prone to cell aggregation, presumably due to manipulation of the pheromone signaling pathway and associated cell polarization¹, adhesion², and fusion³. This phenotype may also contribute to reduced transformation efficiencies, which we have previously observed with similarly engineered strains in our lab. Due to time restrictions, we were unable to fully resolve these issues.

Thus, we regret that we were unable to perform the requested experiment. However, we note that the novel methodology to investigate high-order epistasis between distal sites is our primary message, which we believe is strengthened by our remaining revisions as suggested by the reviewers. In other words, the activation of G-protein signaling in the heterologous yeast strain would be a nice complement to our work, but would not change the overall message of our study. Therefore, we hope that our revisions are sufficient to meet the standards for publication in *Nature Communications*.

At 10 μ M ligand, all possible sites (at least for WT or better binding variants) should be saturated so the selection was mainly done on expression levels. This point is not accurately discussed in the paper.

We thank the reviewer for this comment and agree that our choice of ligand concentration should be clarified in the context of screening for improved ligand binding affinity. In our experiments, the apparent K_d of the fluorescent ligand, FITC-APEC, differs significantly from the literature value reported for A_2aR suspended in bovine striatal membranes, 57 nM⁴. We attribute these differences to transport limitations presented by the yeast cell wall, which has been addressed previously through permeabilization of the cell wall⁵. In our screen, 10 μ M FITC-APEC is near-saturating (**Supplementary Fig. 1 and Supplementary Fig. 4**), which we reasoned would promote enrichment of variants exhibiting gains in either binding affinity or functional yield. Given the rarity of beneficial mutations in sequence space, we reasoned that the choice of this ligand concentration to screen our library would provide useful information in the case that variants with improvements in binding affinity are scarce. We now address these views in the manuscript as shown below:

On lines 137 – 143, we now state,

“Notably, the binding affinity for FITC-APEC is significantly less than that measured for A_2aR suspended in bovine striatal membranes⁴. The yeast cell wall has been reported to hinder transport of fluorescent ligands to the plasma membrane, necessitating permeabilization to facilitate saturation of membrane-embedded receptors⁵. Our results suggest that the use of high FITC-APEC concentrations leads to effective saturation without cell wall permeabilization. Further, saturation promotes enrichment of variants with improved functional yields; thus, cell wall modification is unnecessary to provide conditions favoring enrichment of variants with improved binding affinities.”

On lines 173 – 176, we state,

“We chose to use a near-saturating ligand concentration for library screening after taking into consideration the rarity of beneficial mutations. We reasoned that this strategy would enable enrichment of variants with gains in functional yield, which would provide useful data, if variants with gains in binding affinity are scarce.”

A more detailed explanation of the BRIDGE method (for the non-expert in NGS technologies) would help.

We agree with the reviewer that a detailed explanation of BRIDGE would be valuable, particularly for readers without expertise in next-generation sequencing. Accordingly, the following text has been added to the supplemental information as Supplementary Note 1 to provide details regarding the BRIDGE methodology. The supplemental note is referenced in the main manuscript in lines 181 – 183,

“This processing step leverages the bridge amplification mechanism utilized during paired-end sequencing for Illumina SBS platforms³⁷ (see detailed explanation of BRIDGE in **Supplemental Note 1**).”

“Supplemental Note 1: Detailed explanation of BRIDGE methodology

Below, we provide a detailed explanation of the BRIDGE methodology:

Illumina next-generation sequencing (NGS) platforms are almost exclusively used for deep mutational scanning (DMS) studies due to their high accuracy and throughput⁶. Illumina NGS platforms employ sequencing-by-synthesis (SBS) technology wherein an engineered DNA polymerase replicates a DNA strand in the presence of fluorescently-labeled dNTPs with modified 3' hydroxyl groups, which enable step-wise addition and identification of dNTPs⁷. Upon incorporation of a labeled dNTP, an optical system detects the identity of the nucleotide, then fluorophore release and 3'-modification is catalyzed allowing the next labeled dNTP to be incorporated. The resulting sequence is called a read, which can reach up to 300 bases in length using current technology. Parallelization is achieved through the use of a sequencing chip surface modified with two unique DNA adapters, which bind DNA strands carrying complementary, flanking sequences that can be appended to any strand through PCR (**Supplementary Fig. 13**).

Supplementary Fig. 13: Illustration of DNA template attachment and bridge amplification on Illumina sequencing chip surface. (a) Illumina sequencing platforms feature two unique DNA adapters, which complement flanking sequences appended to a DNA strand (depicted in red and blue). Following interaction of complementary sequences in the strand and on the surface, the distal end of the bound strand approaches the surface facilitating binding to the second adapter. This second interaction enables bridge amplification, wherein the strand is used as a template for DNA synthesis from the second adapter.

(b) Repeated cycles of bridge amplification produce clusters of identical DNA template strands covalently linked to the sequencing chip surface.

In a process referred to as bridge amplification, the distal ends of bound DNA strands approach the chip surface facilitating interaction between the complementary sequences in the template and on the surface. The DNA template is replicated from the second adapter generating a covalently-bound DNA strand. Accordingly, millions of unique DNA strands adhere to the surface and are clonally amplified through repeated rounds of bridge amplification to generate clusters comprising thousands of identical, proximal copies, significantly improving the signal-to-noise ratio⁸. Bridge amplification also enables paired-end sequencing, wherein a single DNA strand is sequenced from both ends providing additional sequence information or improving accuracy if the paired-end reads overlap (**Fig. 1b**). However, due to limitations in read length (i.e. ≤ 300 bases), paired-end reads will only overlap if the 5'-ends of each read are separated by < 600 bases. Thus, distal mutations separated by ≥ 600 bases preclude sequencing using overlapping paired-end reads.

In addition to reducing sequencing accuracy, non-overlapping paired-end reads preclude the use of available DMS software to match reads arising from the same DNA strand. Identification of all mutations arising in a single gene is crucial due to the poorly predicted effects of epistasis. We provide a straightforward method to extend DMS to protein libraries containing distal mutations by leveraging the proximity with which paired-end reads generate fluorescent signals upon incorporating nucleotides into the synthesized read (**Fig. 1b**). For a given DNA strand, the fluorescent signals emitted from each paired-end read share an (x, y) coordinate on the sequencing chip surface. As spatial coordinates are provided for each read in the output FASTQ files, we developed code to parse through the forward and reverse FASTQ read files and match reads that share an (x, y) coordinate.

Following cluster generation, fluorescent signals originating from distinct clusters may overlap on the chip surface reducing the accuracy of matching paired-end reads emerging from the same DNA strand. The occurrence of these polyclonal clusters can be reduced by optimizing clustering density, which can be controlled by the concentration of DNA loaded into the instrument flow cell. However, raw sequencing data passes through a “chastity” or “purity” filtering step in which instrument software automatically discards reads with overlapping fluorescent signals. Accordingly, the sequences contained in FASTQ files are likely to represent those originating from monoclonal clusters.”

Either by chance or design, the five mutagenesis positions were grouped (XX- 750 bp gap-XXX), therefore creating the simplest case scenario for the method. A brief discussion of how it can be extended to the case of three separate groups of mutations would be helpful.

We appreciate the reviewer’s keen observation. In our experiment the library design was guided by available crystal structures and experimental data present in the literature. Additionally, this design facilitated library construction through PCR-driven overlap extension; however, we appreciate the importance of extending BRIDGE to multiple loci. Accordingly, we have provided additional text in the supplement to address this point. We have also modified the README.md file, which now contains detailed instructions to apply BRIDGE to a library containing mutations in three distal loci. Supplemental Note 2 is referenced in the main manuscript in lines 188 – 190,

“Although the mutated sites in our library occupy two distal loci, BRIDGE can be applied to libraries containing mutations in multiple distal sites (**Supplemental Note 2**).”

“**Supplemental Note 2: Application of BRIDGE to libraries containing >2 distal loci**

Although our library was designed to group all mutations in two distal loci, BRIDGE is not limited by positional constraints within individual reads. The mutations may be located anywhere within the length of either read; however, these locations should be chosen taking into consideration the practical considerations noted in **Supplemental Note 3**. As an example, we illustrate below the application of BRIDGE to a protein library wherein mutations are placed in three separate loci (**Supplementary Fig. 14**).

Supplementary Fig. 14: Application of BRIDGE to a protein library containing three distal loci. (a) BRIDGE can be applied to match distal mutations placed in multiple loci as long as each locus falls within either the forward or reverse read. As an example, we describe use of BRIDGE to analyze an A₂A_R library containing mutations at positions T88^{3,36} and Q89^{3,37} (locus 1), L249^{6,51} and H250^{6,52} (locus 2), and S281^{7,46} (locus 3). (b) During paired-end sequencing, the reverse read (step 3) captures the nucleotide sequence at both locus 2 and 3. The forward and reverse reads will share an (x,y) coordinate on the sequencing chip surface enabling *in silico* matching of non-overlapping reads. (c) Practical considerations (e.g. sequencing accuracy as a function of read length) should be made for each library. The library design discussed in this example would place locus 3 near the 3' end of the reverse read, where sequencing errors are more abundant.

Using the Illumina NextSeq sequencer, read lengths reach up to 150 bases enabling use of a reverse read that captures mutations imparted at L249^{6,51}, H250^{6,52}, and S281^{7,46}, which resides on transmembrane helix 7 and has been reported to alter ligand binding affinity upon mutagenesis^{9,10}. Standard protocols

provided by the manufacturer can be used to prepare and sequence the NGS library. Upon obtaining forward and reverse FASTQ read files, BRIDGE can be applied to match non-overlapping paired-end reads generated from the same DNA strand.

The code utilized to match non-overlapping paired-end reads (i.e. `append_reads.py`) does not have to be modified as it relies solely on flow cell coordinates encoded within the FASTQ sequence identifier and is sequence-agnostic. To trim and quality-filter reads using code applied in this manuscript, the `trim-qc()` function can be modified as noted in the README.md file. Upon matching and appending paired-end reads, analysis of enrichment rates may be performed using available software (e.g. Enrich2), or the code described in this manuscript can be modified accordingly. Specifically, appended reads can be translated to their corresponding amino acid sequences by modifying and applying the `translate_reads()` function.”

Figure 3c and Table 1 seem to duplicate each other.

While overall enrichment rates are shown in both Figure 3c and Table 1, we believe that Table 1 is useful in providing a clear rank-order of variant enrichment rates. Specifically, Figure 3c includes both overall enrichment rates as well as those of individual amino acid substitutions. Thus, although Table 1 may serve only to recapitulate the overall enrichment rates, we feel it provides a straightforward presentation of data that may be more difficult to parse from Figure 3c alone. Yet, we value the reviewer’s comment and defer to the editor regarding the inclusion of Table 1 or its relegation to Supplementary Information.

Figure 4 shows a comparable level of expression as measured by radioligand binding (B_{max}) (ca 70% of the WT) but in the text the authors state 17-fold decrease. It is not clear which data support this statement.

We thank the reviewer for highlighting this statement, which we now realize is worded in a confusing manner. The 17-fold decrease refers to the count of variant GRIA in the PS4 NGS dataset rather than the functional receptor yield (B_{max}) determined through radioligand binding. To clarify this statement, we have modified the text accordingly taking into account the newly obtained radioligand binding data, including variant SLNIG, presented in Figure 4:

On lines 322 – 327, we now state,

“While variants GRIA, SLNIG, and Q89S^{3,37} bind ³H-NECA with similar affinities, GRIA and SLNIG are present in the PS4 population with approximately 17- and 12-fold lower abundance, respectively. Thus, our identification of the variants GRIA and SLNIG underscores the sensitivity of our screen and the utility of our analytical method to directly account for epistasis in a search for protein fitness optima.”

Why was the most enriched variant SLNIG not included in the ligand binding assays?

Due to the low throughput nature of radioligand binding assays, we sought to limit the number of variants included in the initial experiment. In addition to controls (i.e. empty vector, wildtype A_{2a}R, and variant Q89A), we sought to investigate a variant in high abundance (Q89S) and a highly-enriched variant. The highly-enriched variant was chosen according to the PS4/Naïve enrichment rate as well as the Enrichment Score calculated using Enrich2 software (Supplementary Table 5), which is commonly used to assess deep mutational scanning data. The main difference between our calculation and that performed using Enrich2 is that Enrich2 calculates an enrichment score based on a linear regression of a variant’s enrichment across all rounds of screening. In contrast, our enrichment rate was calculated by dividing the final abundance of a variant by its initial abundance, which is also a common metric used in deep mutational scanning literature. Although variant SLNIG was the most highly enriched from the naïve to

the final (PS4) population, its Enrich2 score was less than that of variant GRIA, which had the highest Enrich2 score.

Nevertheless, we repeated the radioligand binding assay including variant SLNIG. As shown in the new results presented in Figure 4, below and on pg. #, variant SLNIG exhibits a 3.7-fold improvement in ligand binding affinity relative to the wildtype receptor. The functional yield of SLNIG is improved slightly, approximately 2-fold compared to wildtype. These results support BRIDGE as an effective method to take into account high-order epistasis in the search for variants with improved binding affinities. Given these new results, we have modified the text as follows:

On lines 302 – 313, we now state,

“In line with our goal, all assayed variants bind ^3H -NECA with approximately 4-fold greater affinities than the wildtype receptor, which exhibits an equilibrium dissociation constant (K_d) of 39.2 ± 4.8 nM in agreement with a previous report¹¹ (Fig. 4). The highly-enriched variants, GRIA (K_d of 10.9 ± 0.8 nM) and SLNIG (K_d of 10.7 ± 0.6 nM), bind the target ligand with affinities similar to those of variants Q89A^{3,37} (K_d of 9.9 ± 0.4 nM) and Q89S^{3,37} (K_d of 11.8 ± 1.0 nM). Although these results suggest that differences in enrichment rates do not necessarily correlate with changes in ligand binding affinity, improvements in fluorescent ligand binding may not directly translate to the binding of ^3H -NECA. Nevertheless, BRIDGE enables identification of highly-improved variants subject to high-order epistasis between distal sites. Variants GRIA and SLNIG represent sequences that would be difficult to discover using conventional methods; thus, their identification and inclusion in functional receptor data sets will be useful in informing studies on GPCR structure-function relationships.”

Figure 4: Saturation radioligand binding reveals improved ligand binding affinities in highly-enriched A_{2A}R variants. (a, b) Compared to wildtype A_{2A}R, highly enriched variants, Q89A/S, GRIA, and SLNIG bind [³H]-NECA with 4-fold greater affinity (K_d). (c) Functional yields (B_{max}) of the highly enriched variants are 1.2 – 1.7-fold of the wildtype receptor yield suggesting improvements in ligand binding affinity rather than functional yield dictate variant enrichment. Data represent the mean of three biological replicates, and error bars represent their standard deviation.

The is a statement in the text that individual mutations were all detrimental but no data to support this are not shown.

We believe the reviewer may be referring to the text on lines 324 - 326, “It is only specific combinations of residues that result in such pronounced enrichment rates, while in most other contexts (i.e. sequence

backgrounds) an individual residue is either insufficiently beneficial or even deleterious to the variant's enrichment.”

We agree with the reviewer that deleterious substitutions are not explicitly shown as Figure 5 only presents data for variants with enrichment rates ≥ 1 . The design of Figure 5 suffers from the inherent difficulty of presenting the multi-dimensional data that contributes to epistasis. Specifically, each mutated position represents a different dimension such that there are 5 positional dimensions and 1 dimension corresponding to enrichment rate. Thus, we sought to maintain a balance between information density and accessibility to the reader. Indeed, while generating versions of Figure 5, we found that the inclusion of deleterious variants results in significantly increased information density and a convoluted diagram, which is difficult to understand. As a result, we set a threshold enrichment rate of 1 for Figure 5, wherein the nature of individual residues being insufficiently beneficial or even deleterious in a given sequence is inferred from the absence of specific variants in the figure. We attempted to motivate this inference through Figures 5b and 5c, which present contrasting phenotypes.

In Figure 5b, variants SLNIG and TSWIH are connected to few variants that were enriched ≥ 1 -fold from the naïve to the PS4 population. In contrast, Figure 5c shows variant TSWLH, which is connected to a significantly increased number of variants with enrichment rates ≥ 1 . Thus, we posit that the “isolated networks of variants SLNIG and TSWIH underscore the scarcity of paths originating from disconnected functional variants, which suggests that specific combinations of residues lead to pronounced improvements in function rather than a few critical residues.” We also note a caveat to our choice of figure design in that “While functional variants absent in our processed data set may exhibit significantly improved function or provide additional paths to enriched variants, we find that variants with modest enrichment rates are highly-interconnected (Supplementary Fig. 10). Thus, the scarcity of edges and disconnected networks associated with the highly-enriched variants (Fig. 5b) suggest significant contribution of epistasis to variant fitness.”

Taking into consideration the challenge of clearly presenting analyses drawn from such a large data set and our effort to do so, we hesitate to modify Figure 5, but we welcome suggestions to more clearly present the data referred to by the reviewer.

The quality of the ligand binding data shown does not allow the accurate determination of the K_d s, especially for the key variant GLIA, on which a lot of subsequent analysis and claims are based. The data clearly suggest that it binds but the quality needs to be improved. As a suggestion, homogeneous fluorescence based binding assays (HTRF and nanoBRET) are available for A2A and are probably superior to the radioligand binding in throughput.

We thank the reviewer for their comment and suggested experiments. Although we lack the infrastructure to conduct HTRF or nanoBRET experiments, we repeated the radioligand binding assay using additional, lower ligand concentrations, which provide greater accuracy in K_d determination, particularly for variant GRIA. As shown in Figure 4, we corroborate our previous results, which indicate variant GRIA exhibits improved ligand binding affinity compared to wildtype. The new data also suggest that the K_d of GRIA is comparable to those of variants Q89A, Q89S, and the newly assayed variant, SLNIG. Although these results may suggest that enrichment rate does not strongly predict improvements in ligand binding affinity, we note that differences between the fluorescent ligand used for FACS and ^3H -NECA used for radioligand binding may contribute to these observations. Additionally, we emphasize that GRIA and SLNIG are indeed variants that exhibit improved ligand binding affinity compared to wildtype. Accordingly, the newly obtained data remains in strong support of BRIDGE as a straightforward method to account for high-order epistasis between distal mutations. In light of these new observations, we have modified the text on pgs. 302 - 313 as described in a previous response to reviewer and as shown below:

“In line with our goal, all assayed variants bind ^3H -NECA with approximately 4-fold greater affinities than the wildtype receptor, which exhibits an equilibrium dissociation constant (K_d) of 39.2 ± 4.8 nM in agreement with a previous report¹¹ (Fig. 4). The highly-enriched variants, GRIA (K_d of 10.9 ± 0.8 nM) and SLNIG (K_d of 10.7 ± 0.6 nM), bind the target ligand with affinities similar to those of variants Q89A^{3,37} (K_d of 9.9 ± 0.4 nM) and Q89S^{3,37} (K_d of 11.8 ± 1.0 nM). Although these results suggest that differences in enrichment rates do not necessarily correlate with changes in ligand binding affinity, improvements in fluorescent ligand binding may not directly translate to the binding of ^3H -NECA. Nevertheless, BRIDGE enables identification of highly-improved variants subject to high-order epistasis between distal sites. Variants GRIA and SLNIG represent sequences that would be difficult to discover using conventional methods; thus, their identification and inclusion in functional receptor data sets will be useful in informing studies on GPCR structure-function relationships.”

Figure 4: Saturation radioligand binding reveals improved ligand binding affinities in highly-enriched $A_{2a}R$ variants. (a, b) Compared to wildtype $A_{2a}R$, highly enriched variants, Q89A/S, GRIA, and SLNIG bind [^3H]-NECA with 4-fold greater affinity (K_d). (c) Functional yields (B_{max}) of the highly enriched variants are 1.2 – 1.7-fold of the wildtype receptor yield suggesting improvements in ligand binding affinity rather than functional yield dictate variant enrichment. Data represent the mean of three biological replicates, and error bars represent their standard deviation.

Additionally, it is not clear from the legend if the data shown represent total binding or if they were corrected for the non-specific binding (although it says so in the methods).

We thank the reviewer for pointing out this detail. We have modified the y-axis title in Figure 4a from “counts per minute” to “specific counts per minute” in order to reflect our methods, which include correction for non-specific binding.

In the discussion, there is a statement that the enrichment for some sequences was up to 2,600-fold. However, no such data is shown.

The 2600-fold enrichment rate for variant SLNIG was calculated from the enrichment rates of SLNIG (791) presented in Figure 3c and Table 1 and of WT (0.3) presented in Supplementary Table 2. However, we agree with the reviewer’s sentiment that the data is not directly shown, and that the statement would benefit from a clearer presentation of the data. Therefore, we have added a column to Table 1, which displays Fold Increase in Enrichment Rate over WT.

Again, we would like to thank the reviewer for their careful consideration of our manuscript. We hope that the additional experiments and revisions satisfy the reviewer's suggestions and that our manuscript meets the high standards required for publication in *Nature Communications*.

Reviewer #2 (Remarks to the Author):

Naturally evolved proteins are typically metastable, and most random mutations destabilize native protein structures. For these reasons, evolutionary pathways are highly constrained, and many sequences with useful activities are inaccessible. Next generation sequencing tools have given rise to new experimental approaches including deep mutational scanning (DMS), which are providing unprecedented insights into the mechanisms of protein evolution. DMS technologies have also opened new doors for protein design. In this work, Yoo et al. develop a novel sequencing technique that facilitates facile DMS of sequence-distant sites using short, paired-end reads on conventional Illumina sequencing platforms. They then apply this approach to evaluate the effects of higher order epistatic interactions on the binding affinity of the adenosine A2a receptor, which is an important G-protein coupled receptor. Their results identify new mutant receptors that significantly increase the binding affinity of the receptor for a specific ligand, and illuminate the evolutionary barriers to the sampling these specific variants. This work is generally of very high quality and originality, and offers both methodological advances as well as some fundamental insights. I expect it to have broad relevance to several biochemical and biophysical research communities. Nevertheless, I have several suggestions to improve the manuscript:

We thank the Reviewer for carefully reviewing our paper and making helpful suggestions to improve its content. Below, we address the Reviewer's suggestions point-by-point:

-The higher order epistasis illuminated herein is interesting, and I think the authors do a particularly good job of explaining these connections graphically. However, the authors do not address the potential epistatic mechanisms at all. Elucidation of epistatic mechanism is perhaps beyond the scope of this investigation. Nevertheless, the authors could at least discuss the potential mechanisms (ie folding versus binding). Based on the collective work of the Tawfik, Arnold, and Bloom groups, we may expect most of these epistatic effects to arise from folding constraints. If this were the case, one might expect most of these single variants that are poorly-tolerated in isolation to compromise expression. This possibility could be easily tested by measuring the B_{max} values for a few of the single variants that are depleted from the pool. A comparison of a handful of select Q89 versus W246 mutations could help to illuminate the reasons why most of these pathways seem to be forbidden- do they express but not bind, or do most of these individual substitutions compromise expression, or both? This exercise would significantly enhance the relevance of these results considering the scarcity of deep mutational scanning data for integral membrane proteins.

We thank the reviewer for this suggestion. The elucidation of epistatic mechanisms observed in this study is enticing; however, we agree with the reviewer that this goal is outside of the scope of our investigation. Namely, we focus on the introduction of a new deep mutational scanning method to directly take into account the effects of high-order epistasis between distal sites. Elucidation of the contributions of various biological phenomena to these epistatic interactions represents substantial subsequent efforts. This effort is particularly challenging as multiple factors may contribute, in varying degrees, to the phenotypes of protein variants.

Nonetheless, we agree with the reviewer that the potential underlying mechanisms should be discussed in the context of our study. Accordingly, we have added the following text to the discussion. Please note that newly added text is the portion that is not italicized.

On lines 433 – 460, we state,

“In our screen, we relax several constraints necessitating only proper folding, trafficking to the plasma membrane, and agonist binding. Thus, while increased tolerance to substitutions may be expected, we observe a small number (740) of variants that exhibit improved agonist binding reflecting the inherent constraints imposed by epistasis that are not revealed through single site mutagenesis. The selective pressures imposed in our screen may also point to putative functional effects of epistasis on variant phenotypes and enrichment rates. In different sequence backgrounds, residues contributing to enrichment may instead promote impaired folding, thermodynamic stability, trafficking, or capacity to bind the fluorescent ligand. The scarcity of permissible sequences can be attributed to the rarity of compatible combinations of functionally beneficial, but often destabilizing, mutations and compensatory, stabilizing mutations¹². For example, the evolution of cortisol specificity in the glucocorticoid receptor was mediated by a rare combination of 7 mutations^{13,14}. Among these 7 mutations, 5 are responsible for altered specificity but render the receptor non-functional in the absence of the 2 remaining stabilizing mutations, which are highly specific due, at least in part, to their recruitment of structural rearrangements that directly compensate for conformational changes induced by the destabilizing mutations. The absence of the stabilizing mutations did not significantly affect receptor expression levels suggesting the conformational change, rather than reduced protein degradation or aggregation, permitted functional cortisol specificity¹⁴. The locations of the mutated sites in the A₂AR binding pocket may also facilitate specific epistatic interactions^{15,16}. In other words, interactions between particular substitutions and the ligand or nearby residues may yield specific conformational changes leading to improved function. However, permissible substitutions are only tolerated in the presence of specific residues that act to stabilize the protein variant or properly coordinate atomic interactions with the ligand. Notably, this mechanism is not mutually exclusive with alternative mechanisms underlying epistasis such as rescuing proper surface expression as observed in the neuraminidase protein in oseltamivir-resistant strains of H1N1 influenza¹⁷. Indeed, our observations may also be affected by non-specific epistatic interactions^{15,16}, which contribute to global effects on protein biophysical properties and are independent of the site of mutation. To conclusively determine the type and strength of epistatic interactions occurring in each variant, additional experiments must be conducted.”

-The sequencing platform described herein is unlikely to be broadly impactful in the long-run given the impending rise of sequencing technologies with longer reads (ie Oxford Nanopore and PacBio). Nevertheless, the author has correctly pointed out that these approaches are somewhat boutique and expensive at the moment. For this reason, the BRIDGE technique described herein will likely prove useful for certain experiments. However, the reader would benefit from some discussion on the practical limitations of this approach. Are there likely to be size/ distance restraints for matching the paired end reads? Can you pair reads that are 500 bp apart (likely), 1 kb, 10 kb? How large can the amplicons get before it becomes difficult to match, or does this matter at all? Does the density of clusters on the cell impact your ability to match reads? These may be considerations that are obvious to the authors, but they may not be so obvious to the casual reader.

We thank the reviewer for this comment. We agree that it is prudent to consider amplicon length limitations prior to using BRIDGE. We have addressed these considerations in a note (Supplemental Note 3) added to the Supplementary Information as shown below. Please note that we introduce the terms bridge amplification and clusters in the newly added Supplemental Note 1 as described in our response to reviewer 1. The Supplemental note is now referenced in lines 478 – 479 in the main manuscript,

“We also note practical consideration in applying BRIDGE to protein libraries (**Supplemental Note 3**).”

“**Supplemental Note 3: Practical considerations in applying BRIDGE.**”

Upon attachment to the sequencing chip surface, DNA templates undergo bridge amplification to facilitate cluster generation and paired-end sequencing (**Supplementary Fig. 13**). Thus, amplification efficiency, which is generally greater for shorter templates¹⁸, is important for successful bridge amplification and sequencing of paired-end reads. Accordingly, paired-end sequencing is often suggested for templates with length ≤ 1 kb¹⁹. However, different libraries may yield varying results. While we predict most templates with length ≤ 1 kb will be compatible with BRIDGE, this methodology can likely be applied to longer templates. Additionally, improvements in sequencing chemistry will likely yield improvements in amplification efficiency for longer templates. Provided successful bridge amplification, BRIDGE can be applied to match paired-end reads produced from the ends of the DNA template (i.e. loci separated by ≤ 1 kb).

Prior to analysis of any template, it is prudent to determine optimal cluster density for the specific DNA template. As the density of clusters on the sequencing chip surface increases, the likelihood of exceeding the optimal cluster density (i.e. overclustering) also increases. Overclustering results in a reduced fluorescence signal-to-noise ratio and increases the difficulty of generating and resolving distinct clusters. Although achieving a cluster density that is less than optimal (i.e. underclustering) retains high data quality, it reduces the amount of data that is obtained from an NGS run. Optimal cluster densities for various Illumina NGS platforms are provided by the manufacturer. The concentration of library DNA leading to optimal cluster density can be determined empirically through instructions provided by the manufacturer.

Note that the efficiency of cluster generation can also be impacted by other variables including nucleotide diversity in the initial cycles. Nucleotide diversity and the efficiency of cluster generation is often increased through the addition of PhiX or 1 – 3 nucleotides in the primers used to generate the NGS library. Manufacturer instructions should be noted in optimizing cluster density.

As noted in Supplemental Note 1, raw sequencing data passes through a “chastity” or “purity” filtering step in which instrument software automatically discards reads with overlapping fluorescent signals. Therefore, the sequences contained in FASTQ files are likely to represent those originating from monoclonal clusters.”

-The authors describe the structural context of the chosen side chains, but it is difficult to visualize without a reference structure. I believe there is at least one nice structure for this receptor, and if not, it should be easy enough to make a homology model of this specific receptor using iTASSER. Either would be useful as a guide for the more structurally-minded readers, and the authors should include a structural figure for reference.

We agree with the reviewer and have added a reference structure, as shown below, for A₂aR bound to the ligand, NECA (PDB 2YDV). The new figure is now referenced in the manuscript on line 154,

“Taken together, the lists of atomic contacts and previously reported mutations were used to identify 5 residues (T88^{3.36}, Q89^{3.37}, W246^{6.48}, L249^{6.51}, H250^{6.52}) (**Supplementary Fig. 5**) for site-saturation mutagenesis to test our GPCR screening pipeline.”

Supplementary Figure 5: Structure of the adenosine A_{2a} receptor bound to NECA. The crystal structure of A_{2a}R bound to NECA (PDB code 2YDV²⁰) is shown including stick representation of side chains belonging to residues investigated in this study (i.e. T88^{3,36}, Q89^{3,37}, W246^{6,48}, L249^{6,51}, H250^{6,52}). The shown structure was generated using a thermostabilized variant, which contains a C-terminal truncation at residue 316 and 4 substitutions including a Q89A mutation. Red lines connect atoms between the ligand and side chains whose van der Waals radii are separated by ≤ 0.4 Å.

Overall, I enjoyed reading this paper. It is very well written, and most importantly, is of excellent quality. Both the methodological advances and basic discoveries are likely to be broadly relevant to those interested in protein evolution and design.

We thank the reviewer again for their helpful comments. We hope that the additional discussion and supplementary figure satisfies the reviewer's suggestions and that our manuscript meets the high standards required for publication in *Nature Communications*.

References

1. Madden, K. & Snyder, M. Cell Polarity and Morphogenesis in Budding Yeast. *Annu. Rev. Microbiol.* **52**, 687–744 (1998).
2. Guo, B., Styles, C. A., Feng, Q. & Fink, G. R. A *Saccharomyces* gene family involved in invasive growth, cell-cell adhesion, and mating. *Proc. Natl. Acad. Sci. U. S. A.* **97**, 12158–12163 (2000).
3. Elion, E. A., Trueheart, J. & Fink, G. R. Fus2 localizes near the site of cell fusion and is required for both cell fusion and nuclear alignment during zygote formation. *J. Cell Biol.* **130**, 1283–1296 (1995).
4. McCabe, T. T., Skonick, P. & Jacobson, K. A. FITC-APEC: A Fluorescent Ligand For A2a - Adenosine Receptors. *J. Fluoresc.* **2**, 217–223 (1992).
5. Schütz, M. *et al.* Directed evolution of G protein-coupled receptors in yeast for higher functional production in eukaryotic expression hosts. *Sci. Rep.* **6**, 21508 (2016).
6. Gupta, K. & Varadarajan, R. Insights into protein structure, stability and function from saturation mutagenesis. *Curr. Opin. Struct. Biol.* **50**, 117–125 (2018).
7. Metzker, M. L. Sequencing technologies - the next generation. *Nat. Rev. Genet.* **11**, 31–46 (2010).
8. Kircher, M. & Kelso, J. High-throughput DNA sequencing - Concepts and limitations. *BioEssays* **32**, 524–536 (2010).
9. Kim, J., Wess, J., Michiel van Rhee, A., Schoneberg, T. & Jacobson, K. A. Site-directed Mutagenesis Identifies Residues Involved in Ligand Recognition in the Human A2a Adenosine Receptor. *J. Biol. Chem.* **270**, 13987–13997 (1995).
10. Jiang, Q. *et al.* Hydrophilic side chains in the third and seventh transmembrane helical domains of human A2a adenosine receptors are required for ligand recognition. *Mol. Pharmacol.* **50**, 512–521 (1996).
11. O'Malley, M. A. *et al.* Progress toward heterologous expression of active G-protein-coupled receptors in *Saccharomyces cerevisiae*: Linking cellular stress response with translocation and trafficking. *Protein Sci.* **18**, 2356–70 (2009).
12. Bloom, J. D., Labthavikul, S. T., Otey, C. R. & Arnold, F. H. Protein stability promotes evolvability. *Proc. Natl. Acad. Sci.* **103**, 5869–5874 (2006).
13. Ortlund, E. A., Bridgham, J. T., Redinbo, M. R. & Thornton, J. W. Crystal Structure of an Ancient Protein: Evolution by Conformational Epistasis. *Science (80-.).* **317**, 1544–1548 (2007).
14. Harms, M. J. & Thornton, J. W. Historical contingency and its biophysical basis in glucocorticoid receptor evolution. *Nature* **512**, 203–207 (2014).
15. Starr, T. N. & Thornton, J. W. Epistasis in protein evolution. *Protein Sci.* **25**, 1204–1218 (2016).
16. Harms, M. J. & Thornton, J. W. Evolutionary biochemistry: Revealing the historical and physical causes of protein properties. *Nat. Rev. Genet.* **14**, 559–571 (2013).
17. Bloom, J. D., Gong, L. I. & Baltimore, D. Permissive secondary mutations enable the evolution of influenza oseltamivir resistance. *Science (80-.).* **328**, 1272–1275 (2010).
18. Head, S. R. *et al.* Library construction for next-generation sequencing: Overviews and challenges. *Biotechniques* **56**, 61–77 (2014).
19. Mardis, E. R. Next-Generation Sequencing Platforms. *Annu. Rev. Anal. Chem.* **6**, 287–303 (2013).
20. Lebon, G. *et al.* Agonist-bound adenosine A2A receptor structures reveal common features of

GPCR activation. *Nature* **474**, 521–525 (2011).

Reviewers' Comments:

Reviewer #1:

Remarks to the Author:

The current version of the manuscript is a significant improvement over the previous submission. The authors addressed most of my suggestions and included extended explanation of the methodology. The quality of radioligand binding experiments is much better – my compliments to the person who repeated them. Perhaps, the only comment remaining is whenever the evolved versions of the receptor retained the ability to activate G proteins. However, I understand the technical difficulties of establishing signalling assays in yeast. I also agree that the activity data would be “nice to have” but they would not affect the conclusions of the paper.

Overall, the revised version is a significant improving and my comments are addressed.

just a very minor suggestion

line 137 significantly lower affinity rather than significantly less affinity

Dmitry Veprintsev

Reviewer #2:

Remarks to the Author:

In my original review, I made several requests. In the following, I have summarized these requests, along with the responses of the authors.

Request 1) Measure B-max for some of the individual variants of interest in order to clarify the nature of the epistatic interactions.

The authors have opted not to carry out additional experiments, and have suggested that this is beyond the scope of this work. They instead added a sentence to clarify their assumptions and constraints. I find this disappointing, but acceptable.

Request 2) Describe in additional detail the constraints of this sequencing approach, including considerations for distance constraints and clustering.

It seems as though the authors have included several detailed supplemental notes in order to clarify these points as well as those raised by reviewer 1. I believe this discussion is quite accessible, and will be helpful for anyone interested in using this approach. I find their response to be adequate.

Request 3) Include a structural figure to help orient the readers with regards to the structural context of these mutations.

The authors have included a structural figure in the revised supplement that illustrates the structural context nicely. I might suggest moving it to the main text. Nevertheless, I find this to be a nice addition to the manuscript, and to address the original request nicely.

Overall, I find these revisions to be adequate. The initial draft was of excellent quality, and these revisions certainly help to round out the manuscript. I have no further requests, and believe that this manuscript exceeds the criteria for publication at Nature Communications.

REVIEWERS' COMMENTS:

Reviewer #1 (Remarks to the Author):

The current version of the manuscript is a significant improvement over the previous submission. The authors addressed most of my suggestions and included extended explanation of the methodology. The quality of radioligand binding experiments is much better – my compliments to the person who repeated them. Perhaps, the only comment remaining is whenever the evolved versions of the receptor retained the ability to activate G proteins. However, I understand the technical difficulties of establishing signalling assays in yeast. I also agree that the activity data would be “nice to have” but they would not affect the conclusions of the paper.

Overall, the revised version is a significant improving and my comments are addressed.
just a very minor suggestion

We thank the reviewer for their critical review of our manuscript, which we feel is much improved following the reviewer's suggested changes.

line 137 significantly lower affinity rather than significantly less affinity

We thank the reviewer for their suggestion, and we have changed the text on line 137 to read,

“Notably, the binding affinity for FITC-APEC is significantly lower ...”

Dmitry Veprintsev

Reviewer #2 (Remarks to the Author):

In my original review, I made several requests. In the following, I have summarized these requests, along with the responses of the authors.

Request 1) Measure B-max for some of the individual variants of interest in order to clarify the nature of the epistatic interactions.

The authors have opted not to carry out additional experiments, and have suggested that this is beyond the scope of this work. They instead added a sentence to clarify their assumptions and constraints. I find this disappointing, but acceptable.

Request 2) Describe in additional detail the constraints of this sequencing approach, including considerations for distance constraints and clustering.

It seems as though the authors have included several detailed supplemental notes in order to clarify these points as well as those raised by reviewer 1. I believe this discussion is quite accessible, and will be helpful for anyone interested in using this approach. I find their response to be adequate.

Request 3) Include a structural figure to help orient the readers with regards to the structural context of these mutations.

The authors have included a structural figure in the revised supplement that illustrates the structural context nicely. I might suggest moving it to the main text. Nevertheless, I find this to be a nice addition to the manuscript, and to address the original request nicely.

Overall, I find these revisions to be adequate. The initial draft was of excellent quality, and these revisions certainly help to round out the manuscript. I have no further requests, and believe that this manuscript exceeds the criteria for publication at Nature Communications.

Although we were unable to complete the requested experiment, we are pleased to hear that the changes that were made are satisfactory to the reviewer. We thank the reviewer again for their review of our manuscript and for their comments and suggested changes.